# Confining donor conformation distributions for efficient thermally activated delayed fluorescence with fast spin-flipping

Weidong Qiu [1], Denghui Liu[1], Mengke Li [1] ✉, Xinyi Cai[1], Zijian Chen[1], Yanmei He[1], Baoyan Liang[2], Xiaomei Peng[1], Zhenyang Qiao[1], Jiting Chen[1], Wei Li[1], Junrong Pu[1], Wentao Xie[1], Zhiheng Wang[1,2], Deli Li[1], Yiyang Gan[1], Yihang Jiao[1], Qing Gu[1] & Shi-Jian Su [1] ✉

Fast spin-flipping is the key to exploit the triplet excitons in thermally activated delayed fluorescence based organic light-emitting diodes toward high efficiency, low efficiency roll-off and long operating lifetime. In common donor-acceptor type thermally activated delayed fluorescence molecules, the distribution of dihedral angles in the film state would have significant influence on the photo-physical properties, which are usually neglected by researches. Herein, we find that the excited state lifetimes of thermally activated delayed fluorescence emitters are subjected to conformation distributions in the host-guest system. Acridine-type flexible donors have a broad conformation distribution or bimodal distribution, in which some conformers feature large singlet-triplet energy gap, leading to long excited state lifetime. Utilization of rigid donors with steric hindrance can restrict the conformation distributions in the film to achieve degenerate singlet and triplet states, which is beneficial to efficient reverse intersystem crossing. Based on this principle, three prototype thermally activated delayed fluorescence emitters with confined conformation distributions are developed, achieving high reverse intersystem crossing rate constants greater than $10^6$ s$^{-1}$, which enable highly efficient solution-processed organic light-emitting diodes with suppressed efficiency roll-off.

Organic light-emitting diodes (OLEDs) have become an ideal candidate for flat planal display and solid-state lighting applications[1]. To utilize the 75% generated triplet excitons in electroluminescence (EL), purely organic thermally activated delayed fluorescence (TADF) materials featuring reduced singlet and triplet energy gap ($\Delta E_{ST}$) were proposed to take the place of the noble metal-containing phosphorescence materials[2]. In the TADF materials, the charge transfer (CT) characters between donors and acceptors can delocalize the fronter orbitals, generating a small $\Delta E_{ST}$[3,4]. And the triplet excitons can upconvert to singlet state to achieve 100% internal quantum efficiency in OLED. Based on this mechanism, multiple OLEDs with very high efficiency can be obtained[5–7]. However, most TADF materials possess relatively long excited state lifetime (generally a few microseconds or longer), which could cause singlet-triplet annihilation and triplet-triplet annihilation in the OLED devices at high brightness, dampening the efficiency and operating lifetime[8–10]. Moreover, TADF can be a sensitizer for fluorescence or multiple resonance TADF (MR-TADF) emitters with narrow band emission, in which a TADF sensitizer with efficient RISC is also urgently

[1]State Key Laboratory of Luminescent Materials and Devices and Institute of Polymer Optoelectronic Materials and Devices, South China University of Technology, Guangzhou 510640, P. R. China. [2]Ji Hua Laboratory, Foshan 528200, P. R. China. ✉e-mail: limk@scut.edu.cn; mssjsu@scut.edu.cn

needed[11–13]. Therefore, understanding the determining factor of the spin-flipping process in practical amorphous film state is the cornerstone for efficient TADF-OLEDs.

As a spin-flipping process, the RISC can be theoretically promoted by increasing spin-orbital matrix element ($H_{SOC}$), reducing the $\Delta E_{ST}$ and introducing multiple intermediate states involved. The $H_{SOC}$ can be promoted by incorporating heavy atoms such as selenium[7,14] and bromine[15]. In addition, exploiting the through-space charge transfer emitters with intrinsic separated frontier orbitals can also reduce the $\Delta E_{ST}$ and boost the RISC process[11,16]. Thorough controlling the donor and acceptor orientation with a triptycene bridge in a through-space TADF emitter, Kaji et al. achieved a remarkable RISC process with rate constant ($k_{RISC}$) of $1.2 \times 10^7$ s$^{-1}$ [16]. In the multiple donors emitters, it was found that the resonance of multiple intermediate states corresponding to a part of molecular structure contributes to the efficient RISC process[17–20]. Moreover, because the CT state generally possess negligible $H_{SOC}$, it was proposed that a three-state model involving a locally excited (LE) state is the key to RISC[21–23]. Adopting the above-mentioned strategies, the excited state lifetime of the TADF emitters can be minimized to a few microseconds or sub-microsecond time-range, corresponding to the rate constant of RISC ($k_{RISC}$) >$10^6$ s$^{-1}$ [11,16,19,24–33].

Despite some progresses have been achieved in developing TADF materials with fast RISC, the previous research scope mainly focuses on the electronic structures and photo-physical properties of isolated molecules such as energy level, LE state, and SOC effect. In actual OLED devices, the emitters are in the amorphous emission layer, where the intermolecular interactions can influence the geometry and dielectric environment, perturbing the single molecular photo-physical behaviors[34–36]. Particularly, when the emitters are in thin film state, the molecules cannot relax into the most stable geometry as that in theoretical calculation or solution state, but a distribution of conformer among the potential energy surface (PES)[37–41]. It was reported that this would lead to the change of the phosphorescence spectra at low temperature[42]. A typical and extreme case for conformation distribution is dual conformation of donor-acceptor (D-A) type emitters, which usually bring about multiple emission characteristics including conventional fluorescence, TADF and room-temperature phosphorescence[43–45]. Nevertheless, the previous investigation on dual conformation pays less attention on the excited state lifetime and RISC process. Moreover, molecular design for efficient TADF materials in terms of conformation distribution has not been proposed. As molecular conformations have a significant influence on the photo-physical properties, concerning the long-lived triplet excitons in TADF OLEDs, comprehending how the conformation distribution in amorphous film state affects the RISC process and excited state lifetime is essential to develop OLEDs with high efficiency and low efficiency roll-off.

Herein, the influence of the conformation distribution on the excited state lifetimes of TADF emitters in amorphous film state is comprehensively investigated by theoretical simulation, photo-physical properties, and OLED device characterizations. We combined a benzene-1,3,5-triyltris(phenylmethanone) (TBP) acceptor[46] previously reported by our group with 1,3,6,8-tetramethyl-9H-carbazole (MCz), 9,9-dimethyl-9,10-dihydroacridine (DMAc) and 10H-spiro[acridine-9,2′-adamantane][45] (aDMAc) donors, namely TBP-3MCz, TBP-DMAc and TBP-3aDMAc respectively (Fig. 1a). The selected molecules have varied donor plane rigidity (TBP-3MCz> TBP-DMAc> TBP-3aDMAc), which is controlled by the bridge moiety of acridine or fused-ring in carbazole. In addition, the multiple donor strategy can also construct multiple radiative transition channels for fluorescence emission, multiple RISC channels for spin-flipping, and intramolecular energy transfer among the multiple conformers[17,46]. According to PES scanning, when forming a D-A structure with TBP, the conformational heterogeneity can be expected due to the distorted donor rings and

steric hindrance, corresponding to dual conformations of TBP-3aDMAc and twisted intramolecular CT of TBP-3MCz and TBP-DMAc. In the doped film state, these molecules exhibit an increasing excited state lifetime in the sequence of TBP-3MCz, TBP-DMAc, and TBP-3aDMAc, which is closely related to the conformation heterogeneity originating from the rigidity of the donors. Further, the conformation distributions of the host-guest systems were simulated by molecular dynamics (MD) and quantum chemistry calculations. The rigid donor shows a confined conformation distribution whereas the flexible donor has a broad conformation distribution in the film state, resulting in a large disorder of $\Delta E_{ST}$ and excited state lifetime. Some conformer distributions with large $\Delta E_{ST}$ would prolong the excited state lifetime in the film state. Guided by these results, a molecular design of adopting the rigid donor with steric hindrance for confined conformation distribution was proposed, and three TADF emitters were developed. They all show short excited state lifetime in the range of 1.3–1.9 μs, and the $k_{RISC}$ were calculated to be greater than $10^6$ s$^{-1}$. As a result, the solution-processed green and blue OLEDs and sensitized MR-TADF OLEDs based on these molecules showed high external quantum efficiency (EQE) >20% along with excellent efficiency roll-off suppression, demonstrating the viability of the confined conformation distribution strategy for efficient OLEDs.

## Results
### Basic photo-physical properties
The target molecules were synthesized via one-step common palladium-catalyzed Buchwald–Hartwig coupling reactions of donors and acceptors in high yield (See methods for details). Firstly, basic photo-physical properties of the molecules in doped film were measured (Fig. 1b, c). 9-(2-(9-Phenyl-9H-carbazol-3-yl) phenyl)−9H-3,9′-bicarbazole (PhCzBCz) was selected as a host material, which has a high triplet energy (2.9 eV), moderate polarity as the conventional carbazole-based host and good solution processibility for high-temperature annealing (the structure was shown in Supplementary Fig. 1a)[47]. All of the TBP-based emitters show a structureless CT emission band, with a peak at 512, 539, and 501 nm for TBP3-3MCz, TBP-DMAc, and TBP-3aDMAc, respectively. For TBP-3MCz with rigid donor, an overlap of fluorescence and phosphorescence spectra (5 ms delayed) was observed, indicating a tiny $\Delta E_{ST}$ (Supplementary Fig. 1b). While for TBP-3aDMAc, the phosphorescence spectrum measured at 77 K shows a LE state character, which is possibly originated from the conformer with large conjunction[42,45], and the $\Delta E_{ST}$ estimated from the peak wavelength is 0.19 eV (Supplementary Fig. 2a). To further investigate the dual conformation characteristic of TBP-3aDMAc, the doping-concentration-dependent PL spectra, and excitation-wavelength-dependent PL spectra were measured (Supplementary Fig. 3). In diluted dichloromethane solution or PMMA matrix with low doping concentration, two emission bands can be observed when changing the excitation wavelength (Supplementary Fig. 3a, b). The short wavelength emission is resembled to the PL spectrum of TBP-3aDMAc single crystal with quasi-axial (QA) conformation, indicating that the short wavelength band is originated from the QA conformer. (Supplementary Fig. 4). When increasing the doping concentration, the shoulder emission vanishes and evolves into the CT emission band at long wavelength from quasi-equatorial (QE) conformer (Supplementary Fig. 3c, d). Moreover, the fluorescence lifetimes of the short wavelength band (450 nm) shorten as increasing doping concentration, indicating efficient Förster energy transfer (FRET) from the QA to QE conformer (Supplementary Fig. 3f). In the 15 wt % TBP-3aDMAc: PhCzBCz system, the energy transfer is complete and only the emission from the QE conformer can be overserved. The variance in donor can also lead to significantly different TADF properties. In the transient PL decays of the doped films, obvious

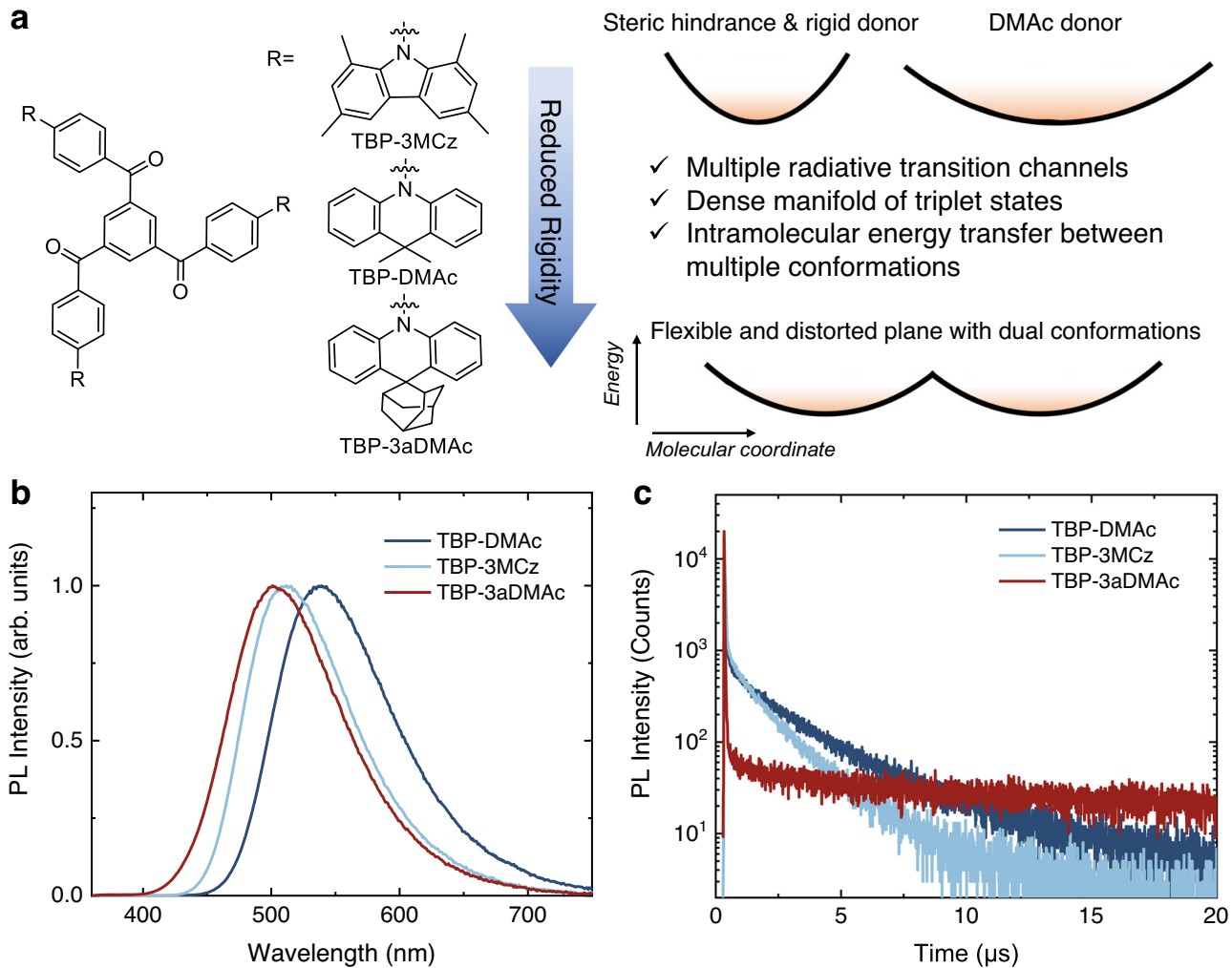

**Fig. 1 | The molecular structures and basic photo-physical properties of the investigated emitters in doped films. a** Chemical structures and characteristics of the multiple-donor-based emitters with different type of donors, and the colored shadings denote the accessible conformations under thermal energy among the potential energy surface. **b** Photoluminescence (PL) spectra of the investigated emitters in doped film. **c** Transient PL decay characters measured in vacuum. Source data are provided as a Source Data file.

**Table 1 | Summary of the photo-physical parameters of the investigated TADF emitters in doped film states**

| | $\lambda_{em}$ nm | $\Phi_{PL}$[a] % | $\Phi_{PF}$[b] % | $\Phi_{DF}$[b] % | $\tau_{PF}$[c] ns | $\tau_{DF}$[c] μs | $k_r$[s][d] $10^7$ s$^{-1}$ | $k_{ISC}$[e] $10^7$ s$^{-1}$ | $k_{RISC}$[f] $10^6$ s$^{-1}$ | $\Delta E_{ST}$[g] meV |
|---|---|---|---|---|---|---|---|---|---|---|
| TBP-3MCz | 512 | 80 | 20.69 | 59.31 | 25.82 | 1.31 | 0.80 | 3.07 | 2.76 | 32.29 |
| TBP-DMAc | 539 | 89 | 40.88 | 48.12 | 23.66 | 3.85 | 1.73 | 2.50 | 0.52 | / |
| TBP-3aDMAc | 501 | 67 | 31.42 | 35.58 | 14.39 | 44.50 | 2.20 | 4.75 | 0.0372 | / |
| TRZ-3MCz | 495 | 89 | 26.53 | 61.47 | 20.02 | 1.63 | 1.33 | 3.67 | 1.94 | 24.58 |
| TB-3MCz | 478 | 83 | 27.63 | 55.37 | 17.20 | 1.94 | 1.61 | 4.21 | 1.43 | 22.16 |

[a]PLQY in $N_2$ condition.
[b]Fraction quantum yield of prompt fluorescence ($\Phi_{PF}$) and delayed fluorescence ($\Phi_{DF}$).
[c]Prompt fluorescence lifetime ($\tau_{PF}$) and delayed fluorescence lifetime ($\tau_{DF}$) from exponential decay fitting.
[d]Rate constant of radiative transition.
[e]Rate constant of ISC, assuming negligible nonradiative transition of singlet state.
[f]Rate constant of RISC, calculated according to Supplementary Methods.
[g]$\Delta E_{ST}$ obtained from Arrhenius fitting of temperature-dependence $k_{RISC}$.

delayed components could be observed, with a lifetime of 1.31, 3.85, and 44.50 μs for TBP3-3MCz, TBP-DMAc, and TBP-3aDMAc, respectively. The photo-physical properties of TBP-DMAc are analogous to the previous report by our group[46]. Notably, with the short delayed lifetime, a large $k_{RISC}$ of $2.76 \times 10^6$ s$^{-1}$ can be calculated for TBP-3MCz. While for TBP-3aDMAc, the dual conformation character is not favorable for the up-conversion of triple excitons, resulting in the lowest $k_{RISC}$ of $3.72 \times 10^4$ s$^{-1}$. The photo-physical parameters are summarized in Table 1. The large difference in the excited state lifetime of these emitters indicates that the conformation of the donor and acceptor plays an important role in the TADF properties.

## Theoretical calculations and flexible potential surfaces scanning

To gain theoretical insight into the excited state properties and conformation energy of these molecules, quantum chemical calculations were conducted. The geometries were optimized by density functional theory (DFT) in B3LYP(D3)/def2-SVP level and the time-dependent DFT (TD-DFT) calculated excited state energies were conducted by range-tuned $\omega$B97XD* functional with def2-SVP basis set to better evaluate the CT character, which match well with the experimental values (Supplementary Fig. 5 and Supplementary Table 2). Both TBP-DMAc and TBP-3MCz exhibit typical twisted intermolecular CT character in the optimized geometries, with D-A dihedral angles of ca. 84° and 78° respectively. Such a large dihedral angle contributes to frontier orbital separations, and results in small $\Delta E_{ST}$ of 0.006 and 0.032 eV for TBP-DMAc and TBP-3MCz, respectively (Supplementary Table 3). Moreover, the multiple donors also lead to triple energetically close degenerated singlet and triplets, which can realize multiple channel radiative transitions and RISC processes (Supplementary Fig. 6)[17, 18]. For TBP-3aDMAc, there would be four possible configurations in one molecule: all QA conformers, two QA and one QE conformers, one QA and two QE conformers, and all QE conformers (Supplementary Fig. 7). From the TD-DFT calculation results, if QA and QE conformers coexist in one molecule, the lowest singlet state will be originated from the twist QE moiety and the lowest triplet state will be originated from the planar QA moiety with large hole and electron overlap. This is in good accordance with the CT-type singlet state and LE-type triplet state of TBP-3aDMAc in spectral characterizations (Supplementary Fig. 2a). In combined with the experimental results, in the film state, the high energy QA state would undergo internal conversion to the lowest singlet state from QE moiety or fast FRET to another molecule with QE conformer. Therefore, fluorescence from low-energy QE conformer is dominated in the 15 wt% doped film. However, as the lowest triplet state lies on the QA conformer, the QA conformation distribution would increase the $\Delta E_{ST}$ values. After ISC or generation of triplet excitons in the EL process, the QA conformer distributions with low triplet state would prolong the excited state lifetime. The calculated $H_{SOC}$ values (summarized in Supplementary Table 3) are quite small in compared with the heavy atom-containing emitters such as selenium[7,14], due to the CT character of the lowest excited state, which is not the main reason for fast RISC. In addition, the LE states of the donor and acceptor moieties have a high triplet energy (>3.0 eV) and large energy gap from $S_1$ state as shown in both theoretical calculation and phosphorescence spectra (Supplementary Fig. 6 and Supplementary Fig. 8), which decreases the possibility for interactions between the CT sate and LE state. Moreover, as the LE and CT state energy relationship can be altered by environment polarity, we measured the transient PL decays of the emitter in hosts with different polarity (Supplementary Figure 9), which shows insignificant changes. This indicates that the LE state is not the most contributing factor in the spin-flipping process in our case. Therefore, we only consider the energy relationship with two-state thermal equilibrium between the $^1$CT and $^3$CT states.

The conformation relationships were first investigated by flexible PES scanning on the D-A dihedral angle. For simplification, the scanned structures possess only one donor connected to the TBP acceptor, which can be representative because it has a similar excited state character and energies with the degenerated states of the structure with multiple donors (Fig. 2a and Supplementary Figure 10). The ground state PESs were calculated by DFT in B3LYP(D3)/6-31G* level, and based on the obtained geometries, the excited state PESs were generated by TD-DFT calculations in the same level of theory (Fig. 2b–d). The ground state PES of TBP-1MCz shows that the conformation energy increases monotonically when the dihedral angle reduces away from the optimized geometry in both ground and excited states, and only one minimum point can be found. The relative percentages of each conformer at room temperature can be calculated according to Boltzmann distribution among the PES (shown in the bar charts in Fig. 2 and Supplementary Table 4). The dihedral angles of TBP-1MCz distributed at the 60°–120° region at room temperature. The similar shape of singlet and triplet PES with close energy indicates that the conformers of TBP-1MCz distribute on the closed singlet and triplet energy region, as revealed by the small averaged $\Delta E_{ST}$ values (23.6 and 25.6 meV for singlet and triple PES respectively). While for the acridine type donor DMAc and aDMAc, two local minima (planar and twist) can be found in ground state. These results are in accordance with the prior research on dual-conformation TADF materials[45,48,49]. And in the excited state, the dihedral angles have a broader distribution than that of TBP-1MCz. In comparison with steeper PES in the singlet state due to the twisted intramolecular CT effect, the triplet state has a flatter PES. This manifests that the molecule would have a broader distribution of dihedral angles in triplet state than in singlet state, resulting in larger average $\Delta E_{ST}$ in triplet state (75.4 meV and 153.8 meV for TBP-1DMAc and TBP-1aDMAc, respectively). Notably, two local minimum conformers with close energy were found in the triplet PES for the dual conformation donor aDMAc, indicating that some TBP-1aDMAc molecules will have planar conformation in triplet excited state according to Boltzmann statistics. Such a planar conformer distribution with large $\Delta E_{ST}$ can be responsible for the significantly prolonged excited state lifetime, and this issue would be more pronounce in the EL process. The averaged $\Delta E_{ST}$ calculated according to PES shows an increasing tendency in the sequence of TBP-1MCz, TBP-1DMAc, and TBP-1aDMAc, which matches their decay lifetimes in the doped films. Consequently, the shape of ground state and excited state PES is controlled by the rigidity and steric hindrance of donors, and finally influences the spin-flipping process.

## Molecular dynamics simulations of the host-guest system

The above discussions were based on the theoretical calculation of single molecule in vacuum, however, in OLED devices, the TADF emitters are in amorphous film state, which involves the varied molecular conformations influenced by intermolecular interactions. In order to explicitly explore the influence of conformation distribution on RISC in host-guest system, MD and TD-DFT simulations were conducted. In the MD calculation, the PhCzBCz host and TBP-3MCz, TBP-DMAc, or TBP-3aDMAc guests were placed into a cubic box with a ratio of 500:55 (Fig. 3a). After thermal annealing and thermal equilibrium, 500 fames were extracted and random 3000 dihedral angles were statistically counted to generate the D-A dihedral angle distributions (Fig. 3b). To estimate the excited state energy distribution in film state, 50 fully equilibrated geometries were extracted from the system and TD-DFT calculations were conducted in B3LYP/6-31G* level of theory and the $\Delta E_{ST}$-oscillator strength relationships were depicted in Fig. 3c. Unlike the DFT optimized result where the conformation is fixed with a settled dihedral angle, the molecules in host-guest systems exhibit a broad distribution from ca. 50° to 90°. This variation might lead to the underestimation of the $\Delta E_{ST}$ by theoretical calculation based on the optimized geometry and prolonged delayed lifetime with multiple exponential decay character in the film state in comparison with that in diluted solution. An extreme example is that the TBP-3aDMAc emitter possesses small $\Delta E_{ST}$ in the quasi-equatorial conformer, however, due to the quasi-axial conformer distribution, the excited state lifetime is very long.

Controlling the conformation distribution for small $\Delta E_{ST}$ is the key to realize efficient TADF with short excited state lifetime. In the TBP-3MCz host-guest system, there is a large average angle of 78.55° and a small standard derivation (SD) of 8.57°. Due to the less rigid donor, the TBP-DMAc system has a smaller average dihedral angle of 75.16° and a larger SD of 8.72° than TBP-3MCz, indicating a broader distribution. Although TBP-DMAc has been proved to be an efficient TADF emitter in OLED devices, the slightly broader distribution of D-A dihedral angle

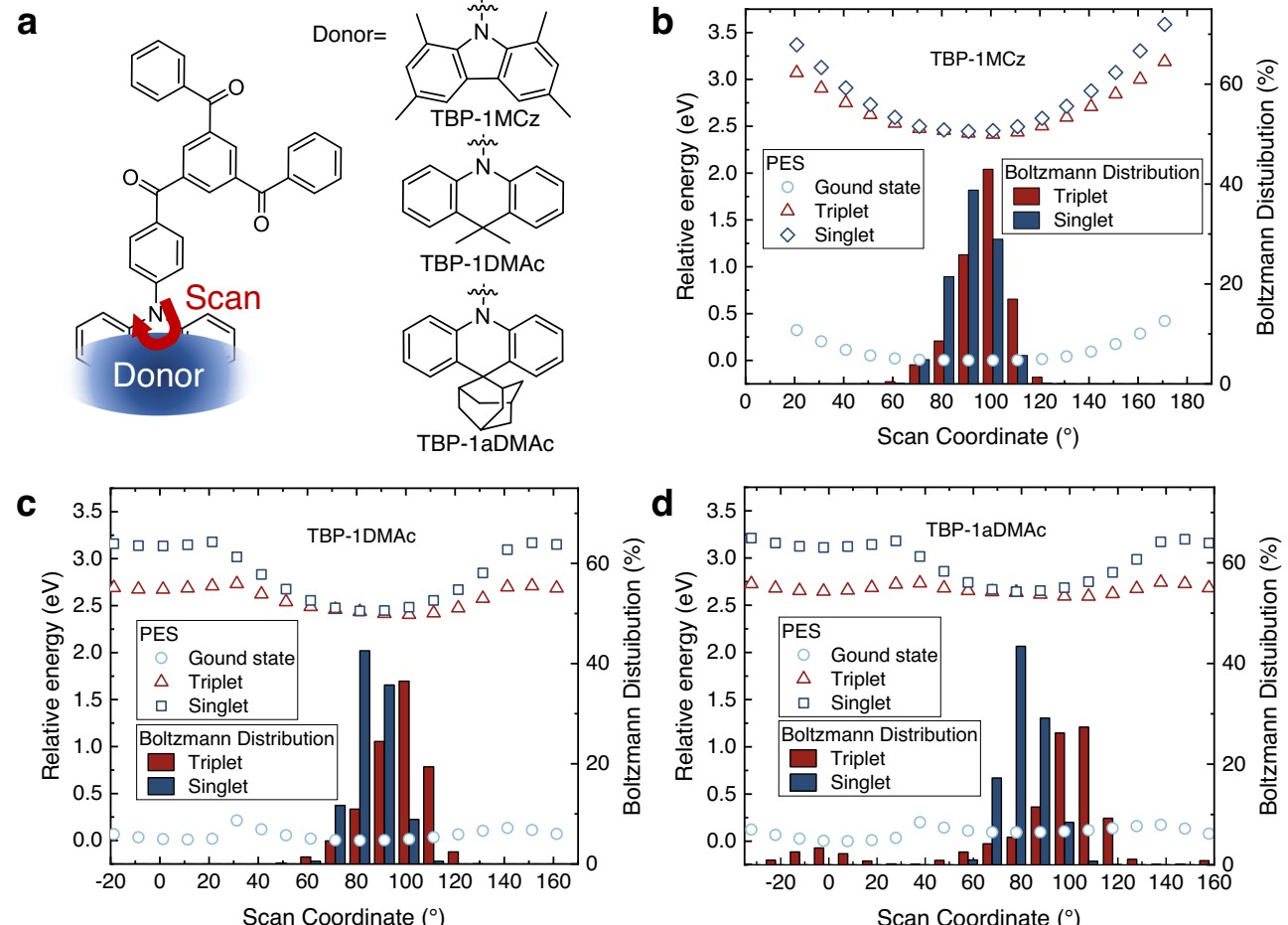

**Fig. 2 | Potential energy surface (PES) scanning of the investigated molecules.**
**a** Illustration of the molecular structures and dihedral angles for PES scanning.
**b**–**d** Flexible PES scanning of the D-A dihedral angles (with single donor) at ground state (light blue symbol) and the correspondence energies in an excited state (red symbol for triplet and blue symbol for singlet state), and the bar charts denote the Boltzmann distributions of the geometries with different dihedral angles in the singlet (blue) and triplet (red) states. Source data are provided as a Source Data file.

originated from the flexible DMAc donor leads to a prolonged excited state lifetime in film state compared with TBP-3MCz. As for TBP-3aDMAc, a clear dual conformation distribution (0–20° and 40–90°) was observed in the simulated host-guest system, resulting in a small average dihedral angle of 60.26° and a very large SD of 25.36°. These results indicate that the rigidity and steric hindrance of the donor in D-A type TADF emitters can regulate the conformation distribution in the host-guest system. The distribution of conformations also leads to disorder of excited state energy (Fig. 3c). The calculated $\Delta E_{ST}$ shows a positive correlation with oscillator strength. This common trade-off relationship between $\Delta E_{ST}$ and oscillator strength in different TADF materials also exists in single emitters with different conformations in host–guest system. TBP-3MCz with confined conformation distribution has concentrated energy distribution (0–0.05 eV), yielding a tiny average $\Delta E_{ST}$ of 0.016 eV and a small SD. In contrast, the $\Delta E_{ST}$ of TBP-DMAc and TBP-3aDMAc show a discrete distribution, and some distributions with large $\Delta E_{ST}$ can also be observed. The larger average $\Delta E_{ST}$ and SD values in the host-guest systems account for the longer excited state lifetimes.

**Time-resolved spectra analyses for conformation distribution**
The conformation distribution can be revealed by time-resolved emission spectra of the doped films experimentally in both PL and EL condition (Fig. 4 and Supplementary Figure 11). The red-shifted spectra in the first 100 ns as delayed time are due to solid state solvent effect

originated from host-guest dipole-dipole interactions. The prompt (0–500 ns) and delayed emissions (>1 μs) were integrated for comparison (Fig. 4a). For TBP-3MCz, both prompt and delayed emissions are almost overlapped with the steady state PL, indicating the less conformation heterogeneity. On the contrary, in the TBP-3aDMAc film, the prompt emission possesses higher energy and larger full-width at half maximum than the delayed emission. This large spectral shift reveals that portion of the molecules having small dihedral angel, including the QA conformation, contribute to the prompt fluorescence. However, these QA conformation distributions cannot utilize the triplet excitons due to the large $\Delta E_{ST}$, which shows the absence of delayed fluorescence. Moreover, as delayed time increases, the delayed fluorescence spectra of TBP-3aDMAc also show conspicuous blue shifts (Fig. 4b). Note that in PhCzBCz host with high triplet energy, the triple excitons can be confined in the guest molecules. Combining the conformation-energy relationship in Fig. 2, the emissions at long delayed time come from the conformers with small dihedral angle (large $\Delta E_{ST}$).

To further confirm the triplet energy distribution effect, we compared the time-resolved PL characteristics of 15 wt%TBP-3aDMAc in CBP host (Supplementary Figure 12). The selection of the CBP host is because it has similar molecular structure and polarity as the PhCzBCz host, but very close triplet state energy to TBP-3aDMAc, which allows for the diffusion of triplet excitons. The 15 wt%TBP-3aDMAc: CBP film shows a double exponential decay with shorter lifetime compared with

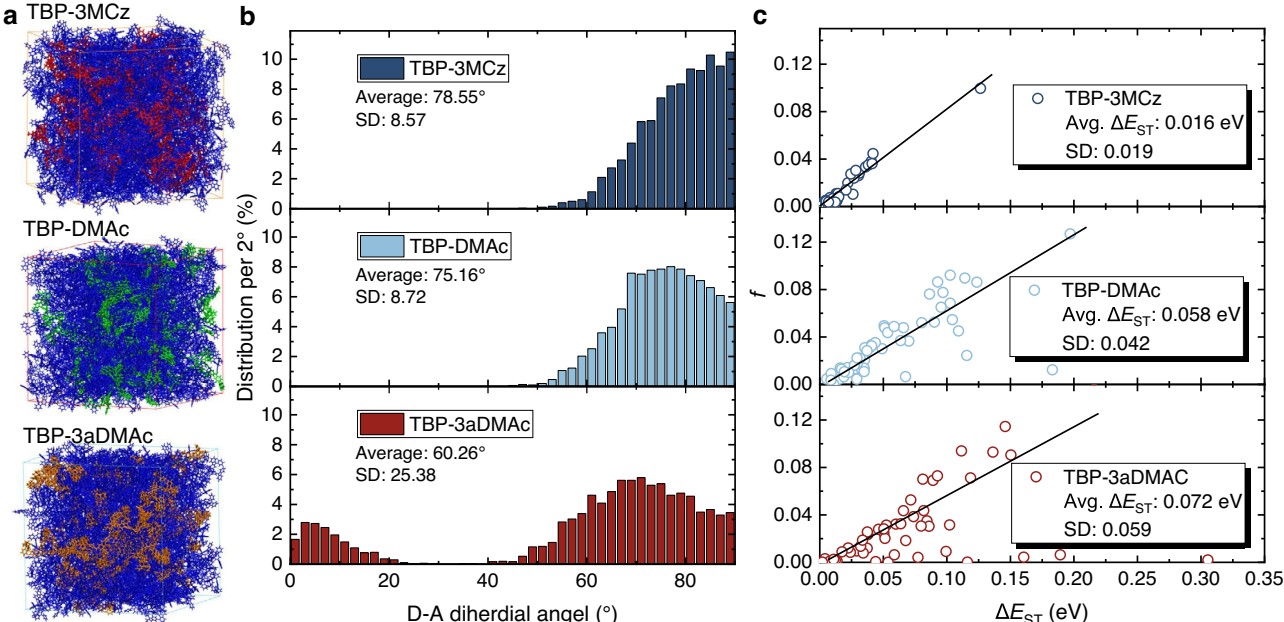

**Fig. 3 | Conformation and excited state energy distributions in host-guest systems from molecular dynamic (MD) and quantum chemistry simulations.** **a** Snapshots of the MD simulation results in thermal equilibrium state. **b** The distributions of the D-A dihedral angles of the investigated molecules in host-guest systems after thermal equilibrium (counting 3000 angles). **c** The TD-DFT calculated $\Delta E_{ST}$ and oscillator strength ($f$) distributions based on 50 geometries generated from MD simulations. Source data are provided as a Source Data file.

the 15 wt% TBP-3aDMAc: PhCzBCz film. Similarly, the prompt emission possesses the short wavelength portion emission while the delayed emission shows a red-shift and narrower spectrum. Notably, the time-resolved PL of the delayed component shows a negligible change as delayed time, which is very similar to that of the 15 wt% TBP-3MCz: PhCzBCz film. Considering the exciton diffusion and quenching process in the low triplet energy host CBP, this can be explained by the triplet excitons in the conformer with large $\Delta E_{ST}$ were quenched or diffused to the conformer with small $\Delta E_{ST}$ to give short TADF, as indicated by the scheme in Supplementary Figure 12e. As for TBP-DMAc, the moderate rigidity of the donor leads to moderate spectral change, which is in accordance with the MD simulations and transient lifetime measurements. Moreover, the conformation distribution in the amorphous film state would differentiate the RISC process in different molecules, causing a distribution of delayed fluorescence lifetime (Supplementary Figure 13 and Supplementary Note). TBP-3MCz with confined conformation distributions results in narrow and short delayed lifetime distributions, while TBP-3aDMAc has a broad lifetime distribution. Combining the energy and lifetime distribution in the doped films, the different conformation distribution effects can be revealed.

In the EL process, the conformation distribution effect would be more pronounce because of the direct generation of the triplet excitons in the flat triplet potential surface. Time-resolved EL spectra of the OLED devices were measured to investigate the conformation distribution in EL condition (Fig. 4c and Supplementary Fig. 14). The transient EL decays are dominated by the delayed component, with increasing lifetime in the sequence of TBP-3MCz, TBP-DMAc, and TBP-3aDMAc. TBP-3aDMAc has a broad conformation distribution in the emission layer. After the carrier injection to generate excitons, the triplet excitons would be accumulated in the conformers with large $\Delta E_{ST}$, forming a plateau with long lifetime in the transient EL decay. In the time-resolved EL spectra, TBP-3aDMAc possesses a more obvious spectral blueshift as delayed time. Estimated from the on-set wavelength, TBP-3MCz, TBP-DMAc, and TBP-3aDMAc show a spectral blue shift of 0.04, 0.05, and 0.08 eV, respectively from the first 1 μs to the

end of emission. Therefore, in the EL process, the conformer distribution with large $\Delta E_{ST}$ is responsible for the long-lived emission, and incorporating rigid donor with steric hindrance can reduce the conformer distribution with large $\Delta E_{ST}$ for short excited state lifetime in OLED.

### Developing TADF emitters with short excited state lifetime

According to the above results, another two TADF emitters with rigid MCz donor were developed (named TB-3MCz and TRZ-3MCz, the molecular structures are depicted in Fig. 5a and Supplementary Fig. 15a). They all process a highly twisted geometry with large dihedral angles, resulting in small $\Delta E_{ST}$ (Supplementary Table 3). In the film state, similar confined conformation distribution can be expected due to the rigidity and steric hindrance of the donors. The photophysical properties of single molecules were investigated in diluted toluene (10⁻⁵ M, Supplementary Fig. 15b, c). All the multiple rigid donor emitters show a CT absorption at 400 nm with large molar extinction coefficient. This verifies that the multiple radiative transition channels induced by multiple donor substitutions can promote fluorescence emission. The fluorescence spectra show a broad and structureless CT emission peaking at 471, 514, and 487 nm for TB-3MCz, TBP-3MCz, and TRZ-3MCz, respectively. And the nearly overlapped fluorescence and phosphorescence spectra measured at 77 K indicate their tiny $\Delta E_{ST}$. Double exponential decay characters were found in these three emitters, and they show longer lifetime after Ar bubbling, indicating the contribution of triplet excitons. Notably, they all show sub-microsecond decay in solution state (delayed lifetime: 0.11–0.14 μs), with which $k_{RISC}$ were calculated to be 8–13 × 10⁶ s⁻¹ (Supplementary Table 5). Although such large $k_{RISC}$ values might be overestimated due to the nonradiative transition, this result indicates that in solution state, the emitters can relax to the optimized geometry for fast spin-flipping, resulting in single exponential fast decay, which is agreed with the tiny $\Delta E_{ST}$ obtained from TD-DFT calculation.

In the 15 wt% PhCzBCz doped film, TB-3MCz and TRZ-3MCz exhibit sky-blue and greenish-blue emission peaks at 478 and 495 nm respectively, with a high photoluminescence quantum yield (PLQY) of

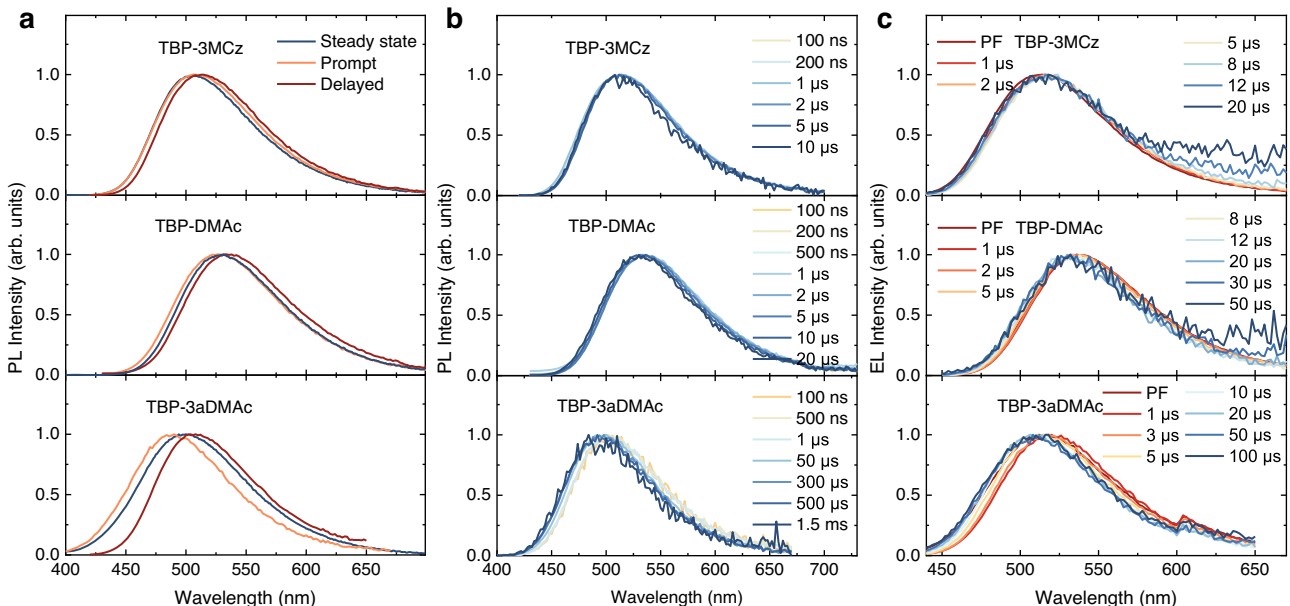

**Fig. 4 | Time-resolved spectra investigations of the doped TBP-3MCz, TBP-DMAc, and TBP-3aDMAc host-guest systems. a** Comparison of the steady-state PL spectra, prompt fluorescence spectra (0–500 ns), and delayed fluorescence spectra (>1 μs). **b** Time-resolved PL spectra of the delayed components after solid-state solvent stabilization and prompt fluorescence emission. **c** Time-resolved EL spectra of the OLED devices based on the host-guest systems (measured with pulse voltage of 6 V and duration of 300 ns), PF denotes prompt fluorescence. Source data are provided as a Source Data file.

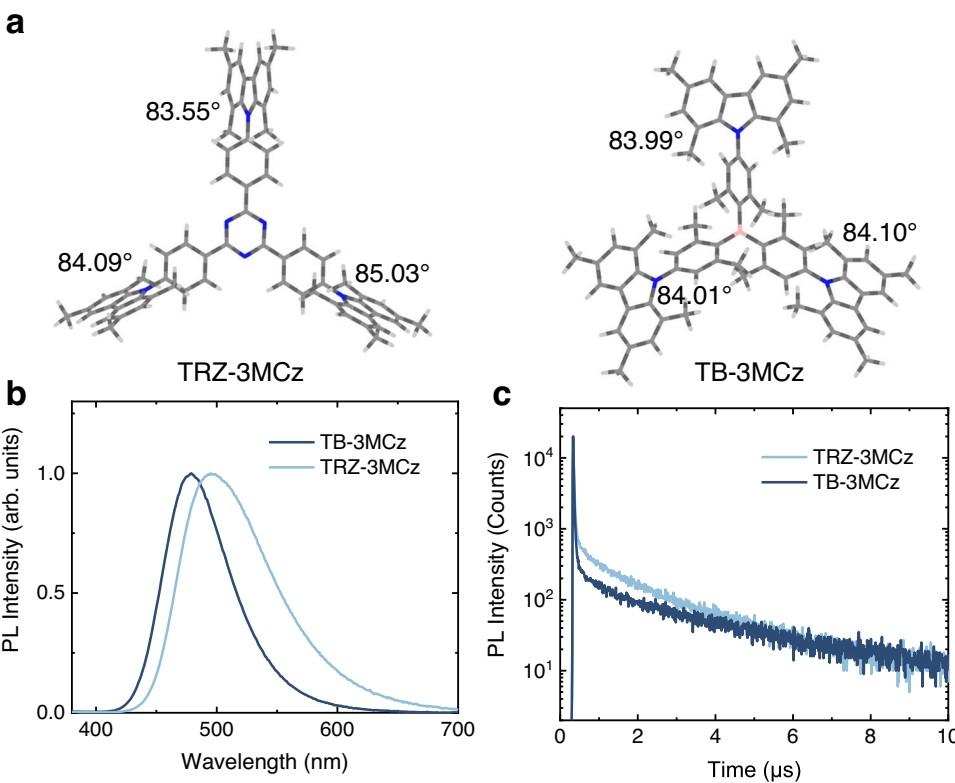

**Fig. 5 | Molecular geometries and basic photo-physical properties of the proposed emitters. a** Optimized ground state geometries of TRZ-3MCz and TB-3MCz. **b** PL and **c** transient PL decay characters of 15 wt% TB-3MCz and TRZ-3MCz doped PhCzBCz films. Source data are provided as a Source Data file.

83% and 89% (Table 1). The fluorescence and phosphorescence spectra of the films have negligible difference, indicating a small $\Delta E_{ST}$ (Supplementary Fig. 16). In compared with the transient PL decay in diluted solution, TB-3MCz and TRZ-3MCz have a prolonged delayed lifetime with multiple exponential decay character in amorphous film state

(1.94 and 1.63 μs, respectively). This can be attributed to the existence of conformer distribution with larger $\Delta E_{ST}$ in the film state than in solution. Nevertheless, with the conformation confinement of MCz donor, TB-3MCz and TRZ-3MCz can also maintain efficient RISC process in the amorphous film state, yielding high $k_{RISC}$ of $1.43 \times 10^6$ and

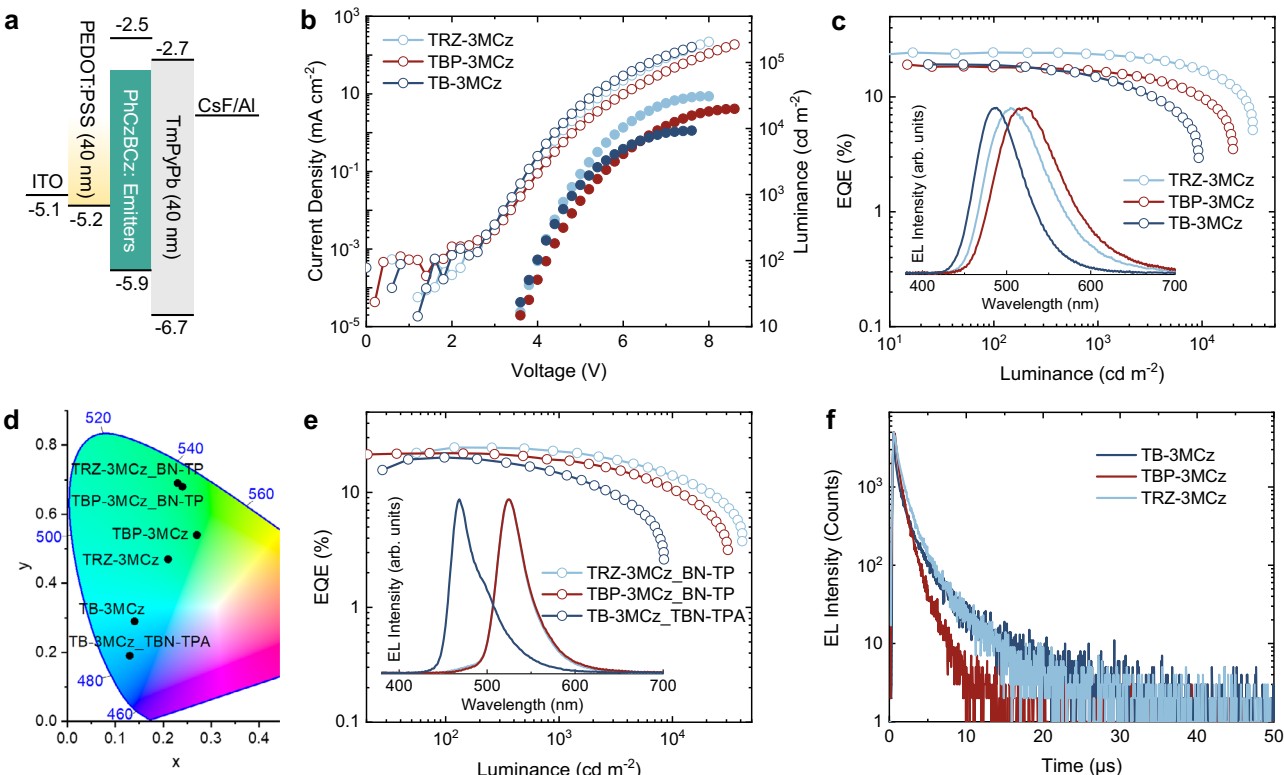

**Fig. 6 | OLED performance of the D-A types emitters with short excited state lifetimes. a** Device architecture and HOMO and LUMO energy levels of the materials in OLED devices. **b** Current-density-voltage-luminance (*J*-*V*-*L*) and **c** EQE-luminance (EQE-*L*) cures of the TADF OLEDs (insert: EL spectra of the devices driven at the current density of 1 mA cm⁻²). **d** Commission Internationale de L'Eclairage (CIE) chromaticity diagram of the EL. **e** EQE-*L* cures of the MR-TADF-based OLEDs with TADF materials as assistant host (insert: EL spectra of the devices driven at the current density of 1 mA cm⁻²). **f** Transient EL decay characters of the TADF OLEDs (measured with pulse voltage of 6 V and duration of 500 ns). Source data are provided as a Source Data file.

$1.94 \times 10^6$ s⁻¹, respectively. According to the temperature-dependent PL and transient PL decay results, the Arrhenius fitting on $k_{RISC}$ can afford very small $\Delta E_{ST}$ values (22–32 meV) for the TADF emitters with rigid MCz donors (Supplementary Fig. 17), which proves the feasibility of the conformation confinement strategy.

To further verify the universality of the strategy, we compared the delayed lifetime and $k_{RISC}$ of a benchmark TADF emitter, DMAc-TRZ, with the emitter with the single or triple MCz substituted triazine in PPF host (Supplementary Fig. 18 and Supplementary Table 6). The delayed lifetime of the 10 wt% DMAc-TRZ: PPF film (2.81 μs) is longer than that of MCz-TRZ[50] with a rigid donor (2.37 μs). When combining multiple donors for multiple radiative transitions and RISC channels with the conformation confinement strategy, the excited state lifetime can be further shortened (1.68 μs in the 10 wt% TRZ-3MCz: PPF film).

**OLED device characterizations**

Given the excellent TADF properties of these materials with confined conformation distribution, solution-processed OLED devices were fabricated to confirm their device performance (Supplementary Fig. 20). The devices were fabricated with a simple three-layer device architecture: ITO/PEDOT:PSS (40 nm)/15 wt% TADF emitter: PhCzBCz (30 nm)/TmPyPb (40 nm)/CsF (1 nm)/Al (Fig. 6a and the molecular structures are shown in Supplementary Fig. 21a). These solution-processed OLED devices show low turn-on voltages of 3.2–3.4 V, and the maximum EQEs of the green emitters TBP-3MCz and TRZ-3MCz reach 19.0% and 24.4%, respectively, while the blue emitter TB-3MCz shows a similar high device efficiency of 19.2%, demonstrating the excellent TADF properties of these emitters in OLED devices (Fig. 6c and Supplementary Table 7). Notably, due to the short excited state lifetimes and high $k_{RISC}$ values, they all show low efficiency roll-off at

high brightness. The best-performing device based on TRZ-3MCz can maintain high EQEs (EQE roll-off) of 23.8% (2.6%) and 21.5% (9.3%) at the brightness of 1000 and 3000 cd m⁻², respectively, representing the best results among the reported solution-processed TADF OLEDs considering the efficiency roll-off and comparable with the vacuum-vaporized OLEDs (Supplementary Table 8)[51]. The device efficiencies of the emitters TB-3MCz and TRZ-3MCz can be further improved to 23.9% and 26.5%, respectively, by inserting a 10 nm PPF blocking layer with higher triplet energy, along with similarly small efficiency roll-off and high maximum brightness (Supplementary Fig. 21). Moreover, vacuum-evaporated OLEDs based on TRZ-3MCz and TB-3MCz show higher maximum EQE of 28.2% and 30.1%, respectively, along with low efficiency roll-off (Supplementary Fig. 22). The improved device efficiencies are mainly due to the horizontal orientation of the transition dipole moments in the vacuum-evaporated films (horizontal ratio of the transition dipole moment of 89%) that increase the light out-coupling, as revealed by the angular-dependent of the emission intensity of the *p*-polarized light (Supplementary Fig. 23). The T₅₀ (the time for EL drops to 50% of the initial 1000 cd m⁻²) for the vacuum-evaporated OLEDs based on TRZ-3MCz and TB-3MCz are 212 hours and 16 hours, respectively.

MR-TADF materials can achieve both high efficiency and narrow-band emission, enabling OLEDs with color gamut covering the most visible region[52–54]. These emitters with fast spin-flipping can also be the assistant host for MR-TADF emitters with narrow-band emission to extend the color gamut. We selected TBN-TPA[55] and BN-TP[56] as blue and green MR-TADF terminal emitters, which possess narrow band emissions and high PLQYs but long excited state lifetimes (Supplementary Fig. 24). Efficient FRET from the TADF assistant host to MR-TADF guest can be found (Supplementary Fig. 25). After doping 1 wt%

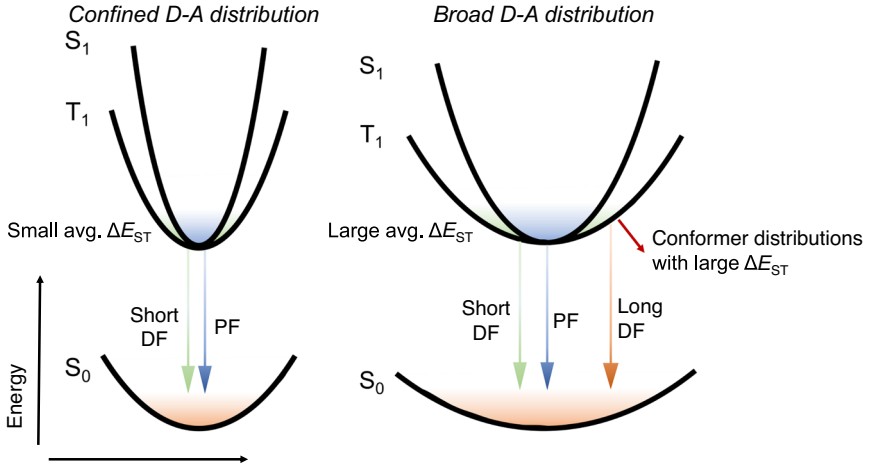

**Fig. 7 | Schematic illustration of the influence of conformation distribution on excited state lifetime of D-A type TADF emitters.** PF and DF denote prompt and delayed fluorescence, respectively, and the colored shadings denote the thermal accessible conformations among the potential energy surfaces of different states.

BN-TP into the emission layer, the green sensitized OLED devices show similar maximum EQEs of 24.6% and 22.0% for TRZ-3MCz and TBP-3MCz assistant hosts with BN-TP as emitter (Fig. 6e, Supplementary Fig. 26 and Supplementary Table 7). And the CIE coordinates of the green OLED devices can be improved to (0.23, 0.69). For the blue-sensitized OLED device, despite the energy transfer being slightly inefficient, a high efficiency of 20.2% can be realized with CIE coordinates improved from (0.14, 0.29) to (0.13, 0.19). Because of the lack of molecular orientation in solution-processed OLED devices (Supplementary Fig. 23), the >20% EQEs of the devices indicate the excellent triplet exciton utilization ability of the TADF emitter or assistant host with conformation confinement strategy.

Finally, we investigated the transient EL decay character of the TADF OLEDs (Fig. 6f). Fast EL decays after the pulse voltage was found in the TBP-3MCz, TRZ-3MCz, and TB-3MCz based OLED devices, with short average lifetimes of 1.18, 1.75, and 2.61 μs, respectively. The emitters with short excited state lifetime can significantly relieve the exciton quenching processes such as triplet-triplet annihilation, triplet-polaron annihilation, and singlet-triple annihilation, resulting in excellent device performances.

## Discussion

From the combination of photophysical property measurements, quantum chemistry calculations, MD simulations of the host-guest systems, and OLED device characterizations, we revealed the crucial role of conformation distribution on the RISC process, which is the key photo-physical process of TADF materials (Fig. 7). Utilization of donors with rigid structure and steric hindrance can strengthen the twisted intramolecular CT effect and steepen the shape of the PES. In this case, similar parabola-shaped singlet and triplet PES with close energy can be obtained, in which the molecules can only distribute among the region with small $\Delta E_{ST}$. For the acridine-type flexible donor with a distorted plane, the PES of D-A dihedral angle can be shallow, and they would have a broad conformation distribution or dual conformation. Moreover, according to the PES scanning results, the triplet state generally has a flat potential surface analogous to the ground state. The conformer distributions with small D-A dihedral angle can lead to a large $\Delta E_{ST}$, prolonging the excited state lifetime. Flexible donor with dual confirmation is an extreme situation for broad conformation distribution, which would significantly prolong the excited state lifetime. The broad conformation distribution effect can be more obvious in the case of EL, where triplet excitons are dominantly generated. Therefore, the conformation distribution in the amorphous film state

of the D-A type TADF emitters is an important factor for the triplet exciton up-conversion process, which should not be ignored when developing novel TADF materials. A feasible strategy is to introduce rigid donors with steric hindrance to confine the conformation distribution. Based on this strategy, in combined with the LE state with large spin-orbital coupling effect or incorporating heavy atom into the emitter might be feasible to further boost the RISC process in the future.

In summary, we revealed the critical role of conformation distribution in host-guest systems on the RISC process in D-A type TADF materials by comparing the photo-physical properties theoretically and experimentally of the TBP-based multiple donor emitters with varied rigidity. The results show that utilizing rigid donors with steric hindrance can restrict the conformation distributions in the film to achieve efficient RISC and short excited state lifetime. Based on this strategy, three proof-of-concept TADF materials were developed, which all show high RISC rate constants greater than $10^6 \, s^{-1}$. Thanks to the short excited state lifetime, highly efficient green and blue solution-processed OLED devices with reduced efficiency roll-off were achieved by utilizing the developed TADF materials as emitters or assistant hosts for MR-TADF emitters. This work extends the scope of TADF materials to conformation disorder in practical amorphous film state in OLED emission layer and provides an effective strategy to regulate conformation distribution toward efficient TADF materials with fast spin-flipping.

## Methods

### Materials

All solvents and reactants were purchased from commercial source (Energy Chemical and Bidepharm). The precursor, TBP-Br (benzene-1,3,5-triyltris((4-bromophenyl)methanone)), TB-Br (tris(4-bromo-2,6-dimethylphenyl)borane) and aDMAc (10H-spiro[acridine-9,2'-adamantane]), were synthesized according to literatures[45,46,57]. TBP-DMAc and MCz-TRZ were synthesized according to literatures[46,50]. Detailed synthesis route and chemical structure characterizations, including $^{1}H$ NMR, $^{13}C$ NMR, and high-resolution mass spectrometry can be found in Supplementary Methods and Supplementary Figs. 27–39. The materials show good thermal stability with thermal decomposition temperatures of 490, 445, and 495 °C for TRZ-3MCz, TB-3MCz, and TBP-3MCz, respectively (Supplementary Fig. 19a). Before OLED device fabrication, the HOMO energy levels were measured to be 5.88, 5.90, and 5.92 eV for TRZ-3MCz, TB-3MCz, and TBP-3MCz, respectively (Supplementary Fig. 19b). The materials for OLED fabrication were purchase from Lumtec Corp. The full name of PPF and TmPyPb is

2,8-Bis(diphenyl-phosphoryl)-dibenzo[b,d]furan and 1,3,5-Tri(m-pyridin-3-ylphenyl)benzene, respectively.

## Photo-physical properties characterization

UV-vis absorption spectra were recorded using a Perkin-Elmer Lambda 950-PKA instrument. Fluorescence and phosphorescence spectra at RT and 77 K were recorded by an FL980 spectrometer (Edinburgh Instrument) equipped with a gated photomultiplier tube. Transient PL decay profiles, time-resolved emission spectra, and temperature-dependent PL spectra were measured on a FL980 spectrometer (Edinburgh Instrument), excited with a 320 nm laser diode. The transient lifetime measurements were conducted in vacuum. For the temperature-dependence measurement, an Oxford Instruments liquid nitrogen cryostat (Optistat DN) was adopted. PLQYs of films were measured by a calibrated integrating sphere (Edinburgh Instrument) excited at 320 nm in FL980.

## Theoretical simulation

Geometry optimizations and flexible potential energy surface scanning were performed by the Gaussian 09 E01 package in B3LYP(D3)/def2-SVP and B3LYP(D3)/6-31G* level, respectively[58]. Based on the ground state geometry, the excited state properties were calculated by TD-DFT calculations. A tuned range-separated hybrid functional $\omega$B97XD* with def2-SVP base set was adopted for the single point excited state energy calculation. The optimal $\omega$ values were around 0.1. A comparison of the calculated energies with different methods can be found in Supplementary Table 2. The hole-electron distribution of excited states were analyzed by the method proposed by Lu et al.[59] using Multiwfn[60] and visualized with VMD software[61]. The corresponding spin-orbital coupling matrix elements between singlet and triplet states were calculated by the embedded PySOC codes in the Molecular Materials Property Prediction Package (MOMAP)[62,63]. All MD simulations were performed using GROMACS software package version 2019.6 with general amber force field (GAFF) force field and restrained electrostatic potential (RESP) atomic charge[64–66]. 500 host molecules and 55 guest molecules were randomly placed into a cubic box with initial size of $15 \times 15 \times 15$ nm$^3$. Conjugated gradient method was used for energy minimization to eliminate the apparent irrational repulsion in the system. After that, a 100-ps equilibrium MD simulation was conducted under NVT ensemble to make the system fully balanced. Then a 10-ns production MD simulation was conducted. Periodic boundary conditions were used in the production MD process. The velocity-rescale temperature coupling with time constant of 0.2 ps and Berendsen pressure coupling with time constant of 0.5 ps were used[67]. Non-bonded interactions were calculated using a cutoff of 14 Å. Configurations used for the dihedral angle statistics and quantum chemistry analyses were extracted from the last 1-ns MD trajectories. The dihedral angles were counted by VMD software. The geometries from MD calculation were extracted randomly from the trajectories by VMD and then transferred to Gaussian for TD-DFT calculation, which can be enabled by Molclus program[68]. The optimized molecular coordinates and MD trajectories can be found in the Source Data file.

## OLED device characterization

Before device fabrication, indium-tin-oxide (ITO) coated glass substrates were washed with ultrasonic sequentially in deionized water, acetone, and ethanol and treated with oxygen plasma for 2 min. PEDOT: PSS (4083, Xi'an p-OLED) was spin-coated onto the plasma-treated ITO substrate at 3000 rpm for 30 s, followed by annealing at 150 °C for 15 min to form a 40 nm hole injection layer. The TADF materials and host were dissolved in chlorobenzene (10 mg mL$^{-1}$). And the MR-TADF emitters can be dissolved in chlorobenzene with a concentration of 1 mg mL$^{-1}$. The emission layer was spin-coated onto the PEDOT: PSS substrate at 2000 rpm for 30 s followed by thermal annealing at 100 °C for 10 min, forming a ca.

30 nm emission layer. The organic electron transport layer TmPyPb and electrode were then deposited onto the substrate in a thermal evaporation system (FS-450, Suzhou Fangsheng) under high vacuum (<10$^{-4}$ Pa). The deposition rates of the organic materials, CsF, and Al were 0.8–1.2 Å s$^{-1}$, 0.1 Å s$^{-1}$, and 3 Å s$^{-1}$, respectively. After fabrication, the devices were immediately encapsulated with a glass cover using epoxy glue in a nitrogen-filled glove box. The current density and luminance versus driving voltage characteristics and EL spectra were recorded by Photo Research PR745 and powered by Keithley 2400. EQE was calculated from the luminance, current density, and EL spectrum, assuming a Lambertian distribution. The size of the pixels is 0.1 cm$^2$. The transient EL and time-resolved EL measurements were conducted on an FL980 spectrometer (Edinburg Instrument), using multi-channel scaling mode for the measurement. The device structures were ITO/PEDOT: PSS (30 nm)/15 wt% TADF emitter: PhCzBCz (30 nm)/TmPyPb (40 nm)/CsF (1 nm)/Al. The pulse voltages were generated by an oscilloscope (RIGOLDG4162), and a −2 V voltage was applied after excitation to eliminate the recombination of free carriers.

## Data availability

The X-ray crystallographic coordinates for TBP-3aDMAc reported in this study have been deposited at the Cambridge Crystallographic Data Centre (CCDC), under deposition number 2191190. These data can be obtained free of charge from The Cambridge Crystallographic Data Centre via www.ccdc.cam.ac.uk/data_request/cif. The spectral measurements, theoretical simulations, and OLED device characterization data generated in this study are provided in the Source Data file. Source data are provided with this paper.

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

## Acknowledgements

The authors greatly appreciate the financial support from the National Key R&D Program of China (2020YFA0714600 to S.-J.S.), the National Natural Science Foundation of China (52273179, 51625301, 91833304, and 51861145301 to S.-J.S.), and the Basic and Applied Basic Research Foundation of Guangdong Province (2019B1515120023 to S.-J.S.). The authors also acknowledge HZWTECH for providing computation facilitates.

## Author contributions

W.Q., M.L., and S.-J.S. conceived the work. W.Q., X.C., B.L., Z.H., Z.Q., J.C., L.W., and Z.W. contributed to the materials involved. W.Q., X.C., J.C., W.L., and D.Li. contributed to the photo-physical properties measurement. W.Q. and Z.C. performed the quantum-chemical calculations. M.L. conducted the MD simulations. D. Liu, Y.H., W.X., X.P., and J.P. contribute to the OLED device fabrication and analysis. W.Q., M.L., Y.G., Y.J., Q.G., and S.-J.S. contributed to the analysis of results. M.L. and S.-J.S. supervised the project. W.Q., M.L., and S.-J.S. wrote the manuscript with contributions from all authors. All authors contributed to the discussion and modification of the manuscript.

## Competing interests

The authors declare no competing interests.
