## [Peer Review File · Nature Communications]

Confining donor conformation distributions for efficient thermally activated delayed fluorescence with fast spin-flippingREVIEWER COMMENTS

Reviewer #1 (Remarks to the Author):

The authors demonstrated using rigid donors can help TADF emitters with fast RISC rate. Although the authors proposed an explanation for the different performances of their developed compounds, they underestimated/ignored other key factors for such complex systems and exaggerated their results. Moreover, the device efficiency is very ordinary. As a result, I don't think this work has the quality and novelty of publishing in Nature Communications. Other major comments are listed below.

1. The authors stated that "the influence of conformation distribution on it has not been explored." However, this statement is not true. In fact, many theoretical studies on this topic have been carried out, for example, PHYSICAL REVIEW MATERIALS 1, 075602 (2017); J. Mater. Chem. C, 2017, 5, 5718; J. Mater. Chem. C, 2017, 5, 11001-11009;
2. I cannot understand the reason why the authors stated "a very large kRISC of $2.76 \times 10^6 \text{ s}^{-1}$ can be calculated for TBP-3MCz." Most of TADF emitters show kRISC values of $> 1 \times 10^6 \text{ s}^{-1}$. Moreover, to date, several excellent TADF emitters were reported to show a kRISC as large as $> 10^7 \text{ s}^{-1}$ (Nat. Photonics 14, 636–642 (2020); Nat. Photonics 14, 643–649 (2020)).
3. I would suggest that the authors used range-tuned ω B97XD* functional in both geometrical optimization and TDDFT calculations.
4. The authors claimed that the conformation-dependent SOCME of TBP-MCz is small, but the variation is significant (Fig. S5). Moreover, the higher triplet states, especially the LE triplet states might play important roles as conformation varies.
5. Line 122, delta EST should be obtained from Supplementary Fig. 2a. Moreover, hole-electron distributions of the triplet states of TBP-3aDMAC should be provided to support their claim of LE character (Line 120).
6. Single crystal results are necessary to support their claims.
7. For TBP-a-DMAC, conformations should be much more complicated than the authors discussed. There should be at least $2 \times 2 \times 2$ types of conformers.
8. What are the distributions of QA and QE conformers in amorphous states? More solid experimental results should be provided rather than only theoretical estimations.
9. In Fig. S6, the authors simply ascribe the variation of time-resolved PL spectra to the solvent effect and conformation heterogeneity. Possible energy transfer between different conformers should be considered. Should the long-tail conformer show a fast energy transfer to the twisted conformer or not? The authors should explain.

10. Have the authors tried to fabricate evaporation-processed OLEDs? Based on the TGA results, I think preparing evaporation-processed OLEDs might be also available.

11. Device operation lifetime should be measured as it is an important claim in the abstract.

Reviewer #2 (Remarks to the Author):

Three new TADF emitters TBP-3MCz, TRZ-3MCz and TB-3MCz have been synthesized, with emphasis on multiple rigid donor components to confine the conformation of the emitters. Highly-twisted geometries give small ST gaps. Fast RISC rates are demonstrated leading to very efficient solution processed OLEDs, with low efficiency roll-off at high brightness. These are currently important topics in the field. This is noteworthy.

The article is logically written. The molecular design is rationally presented. Standard photophysical, structural characterization techniques and device fabrication are used, although the photophysics is rather limited.

The manuscript could potentially become suitable for Nature Comm. But major revision will be required as listed below.

1. Established literature: Conformational control is important in donors and there is discussion in the article of axial, equatorial acridines. Quasi-axial conformer is stated to be TADF inactive. However, this is not a new concept. Established literature is not referenced. Acridine conformers have also been discussed previously, Li, W et al, *Angew Chem Int Ed Engl.* 2019, 58(2):582-586. doi: 10.1002/anie.201811703. This is also well established with phenothiazine donor in D-A molecules: Chen, C et al *Angew. Chem. Int. Ed.* 2018, DOI: 10.1002/anie.201809945.

2. Data analysis. It is stated in SI "Energy transfer from high energy quasi-axial to low energy quasi-equatorial form can happen." This energy transfer between the two conformers should be analyzed in more detail experimentally. Does this depend on the solid-state host? Only PMMA host is reported (manuscript page 4, line 1; SI Fig 2).

3. Data analysis. Emitter layer on OLEDs is 15% doped in PhCzBCz. There should be more discussion on the choice of host and why 15% was chosen. I cannot see data about optimization of dopant concentration.

4. Neat emitter layer, (non-doped) devices could also be reported for comparison.

5. SI. For further unambiguous proof of molecular structures, copies of all mass spec should also be shown as figures in SI.

6. Methodology: Quantum yields. Table 1, Supplementary Table 4 and elsewhere. It is not meaningful to quote photoluminescence quantum yields to one- or two- decimal places level of accuracy, as reported here. This suggests that the authors do not appreciate the inherent inaccuracy in the technique (usually +/- 10% of the value reported). For example, Table 1, line 1, 79.6, 20.59 and 59.01 should be 80, 21 and 59.

7. Methodology: Supplementary Table 5. EQEs cannot be accurate to two decimal places. Please correct this.

Reviewer #3 (Remarks to the Author):

This work examined the influence of the distribution of dihedral angles between donor and acceptor moieties within a TADF emitter on the photophysical properties, such as lifetime, emission profiles. Molecular Dynamic simulations have been used to model the distribution of emitter inside the matrix of host. It was found that emitters that are structurally more rigid showed a narrower conformational distribution, while one emitter showed 2 dominant conformations gave rise to unfavorable longer emission lifetime. However, such effect on TADF behavior is well known in the literature (e.g. J. Phys. Chem. A 2021, 125, 35, 7644)

1. The use of the term “long-tail” seems rather unscientific and sometimes unclear. Would there be other way to describe it?

2. From line 123 onwards, the authors tried to demonstrate the dual conformational character of TBP-3aDMAc by concentration-dependent PL spectra in PMMA and excitation dependent solution emission spectra. According to the authors, increasing doping concentration led to red-shift of emission due to more efficient energy transfer from the quasi-axial to quasi-equatorial conformers. But why would the emission band of the neat film occur at a higher energy than that at 50wt% doping concentration? Could the red shift of emission band be also explained by the increasing polarity of the film with increasing doping concentration? In the solution state, the molecules should have more degree of freedom for bond rotations and therefore would be easier to access other conformations and therefore assignment of only two conformation based on solution state study may not be concrete enough. Due to the highly twisted nature of these molecules, electronic communications between different parts of the molecule may not be efficient resulting in dual emission from just a single conformer. Did the author measure any excitation spectra to exclude such possibility? The vibronic feature in the emission spectra also shows resemblance with the phosphorescent spectra of just the donor part aDMAc. The author may also compute the emission spectra based on the two conformers that have been optimized.

3. Lines 158-160, “ Locally excited states of the donor and acceptor moieties ... can hardly interact with the low-lying CT triplet state...”. Try to tone down a bit.

4. Lines 164-165, “For simplification, only one donor connected to the TBP acceptor was allowed to motion move in the scanning”, does it mean there are two other donor moieties on the other part of the molecule that are fixed, but I don't see those in your structure in Fig 2. The authors need to clarify that.

If there is only one donor is used in your calculation (to reduce computational cost), can it truly reflect the conformational distribution as in the molecules you synthesized?

5. Line 245, What does it mean by solid state solvent effect stabilization? Solvent should be gone at this point, in theory.

6. Line 251, It would be better to also show the TR-spectra of TBP-3MC in thin film state for comparison.

7. Fig 2c and d. Could the author explain why only for the case of TBP-a-DMAc, there is some distribution at angles around 0°, but it is not seen in TBP-DMAc even though they showed very similar energy trends for ground state, singlet and triplet excited states?

8. Inconsistency in naming of compounds have been noted.

9. Lines 469-470, "The transient lifetime measurements were conducted in vacuum" Does the author mean the whole instrument is placed in a vacuum chamber or is the sample after degassing is stored under vacuum?

10. Line 500, does the author use oxygen or ozone to treat ITO glass?

11. Supplementary Line 67, "different conformers have different transition dipole moments"

12. Supplementary synthesis section, Line 239, Pd(OAc)₂ is called palladium(II) acetate.

13. Captions for Supplementary Fig 17-20, ¹³C{¹H} NMR spectra and CDCl₃-d.

14. Could the author explain how the computational methods are chosen since over half of the article is based on computation? Have the authors done any benchmarking?

15. Please provide detailed description on how to measure those rates and also the fitting protocols.

As the explanations offered to explain the experimental findings are not concrete enough and the effect of conformation on TADF behaviour is not unprecedented in the literature, this article may not be suitable to be published in nature communication.

Point-by-point responses to the reviewers' comments:

Reviewer #1 (Remarks to the Author):

The authors demonstrated using rigid donors can help TADF emitters with fast RISC rate. Although the authors proposed an explanation for the different performances of their developed compounds, they underestimated/ignored other key factors for such complex systems and exaggerated their results. Moreover, the device efficiency is very ordinary. As a result, I don't think this work has the quality and novelty of publishing in Nature Communications. Other major comments are listed below.

Response: Thanks for your comments. We did not quite understand what the key factors of the complex system we missed. Based on the comments, we supposed that the previous version of the manuscript has not sufficiently discussed on the LE state with large SOC and the possible energy transfer process between the conformers. They have been investigated in the revised manuscript with additional experiments. Please find them in the point-by-point responses and the main text. Moreover, based on the additional experiments guided by the questions from reviewers, the results and conclusions were strengthened. Therefore, we believe that our results are valid and can be a guideline for developing efficient TADF materials with short excited state lifetime.

With regard to the device efficiency, it is a challenge to fabricate solution-processed TADF OLEDs with low efficiency roll-off, which are seldomly reported. And the reported solution-processed OLEDs with higher EQE values than ours possess severe efficiency roll-off (*Angew. Chem. Int. Ed.* 2022, 61, e202212861; *Adv. Mater.* 2022, 34, 2110344). Based on the proposed strategies, the best solution-processed OLED with TRZ-3MCz as emitter possesses high EQE of 24.4% and can maintain high EQEs (EQE roll-off) of 23.8% (2.6%) and 21.5% (9.3%) at the brightness of 1000 and 3000 cd m⁻², which is the best performance for solution-processed OLEDs considering the efficiency roll-off. We summarized the representative solution-processed OLEDs with conventional device structure in supplementary table 8 for comparison. Moreover, according to the question 11, the vacuum vaporized OLED based on TB-3MCz can achieve a further improved EQE_{max} of 30.1% due to the orientation effect in the vacuum vaporized film along with a reduced EQE roll-off.

Supplementary Table 8. Comparison of the representative solution-processed OLED device performances with conventional device structures.

Compounds	λ_{EL} nm	EQE _{max/500/1000} %	Roll-off _{500/1000} %	Reference
TRZ-3MCz	505	24.4/24.2/23.8	0.8/2.6	This work
TRZ-3MCz	512	26.5/-/26.5	-	This work
TB-3MCz	487	23.9/23.2/20.6	2.9/13.8	This work
TBP-3MCz	521	19.0/17.4/17.0	8.4/10.5	This work
DCz-DPS-TCz	498	24/21.3/-	11.3/-	Angew. Chem. Int. Ed. 2022 , 61, e202115140
tBuCz2m2pTRZ	540	28.7/14.3/-	50.9/-	Adv. Mater. 2022 , 34, 2110344
5CzBN-ESF	508	30.6/23.6/20.5	22.9/33.0	Angew. Chem. Int. Ed. 2022 , 61, e202212861

YD-TF	552	21.9/-/18.6	-/15.1	Angew. Chem. Int. Ed. 2021 , 60,16585– 16593
DMeCzIBN	478	21.6/-/19.6	-/9.3	Chem. Eng. J. 2021 , 412, 128574.
T-CNDF-T-tCz	484	21.0/-/-	-/-	Adv. Sci. 2020 , 7, 1902087.
OAB-ABP-1	506	19.6/-/17.4	-/11.2	Adv. Mater. 2020 , 2004072
TBP-DMAc	526	22.1/-/20.3	-/8.1	Adv. Funct. Mater. 2018 , 28, 1704927

1. The authors stated that “the influence of conformation distribution on it has not been explored.”. However, this statement is not true. In fact, many theoretical studies on this topic have been carried out, for example, PHYSICAL REVIEW MATERIALS 1, 075602 (2017); *J. Mater. Chem. C*, 2017, 5, 5718; *J. Mater. Chem. C*, 2017, 5, 11001-11009;

Response: Thanks for your comments. Conformation-property relationships are significant for developing TADF materials and therefore, they have been investigated heavily in both theoretical simulations and practical device applications. The concerning literatures have been cited in the introduction. The pioneer literature of PHYSICAL REVIEW MATERIALS 1, 075602 (2017) presents the thermal fluctuations of the molecular conformations in amorphous films, which is an important research background of our manuscript. The *J. Mater. Chem. C*, 2017, 5, 5718 investigated the excited state energies, potential energy surface in detail of D-A-D or D-A TADF emitters theoretically. And the *J. Mater. Chem. C*, 2017, 5, 11001-11009 investigated the solid-state solvation effect and explain the observed time-dependent spectral changing effect in ns time scale. Although some researches on a topic have been conducted, there should be additional in deep researches to be conducted and published. For example, *Nat. Mater.* 2022, 21, 1150 investigated the same solid-state solvation effect as *J. Mater. Chem. C*, 2017, 5, 11001-11009 but in different aspect. Nevertheless, the mentioned literatures focus on different aspects in the TADF materials from ours, and with different goals. Our aim is to investigate the effect of conformation distribution on the key RISC process of TADF materials and provide a structure-property relationship for new materials design and OLED applications. Some similar characterization methods such as theoretical calculations were used in our work, but we don't think it weakens the novelty. Moreover, our research is more than theoretical investigation. Based on the conclusion from the theoretical and photo-physical investigations, TADF materials with short excited state lifetimes for efficient OLEDs and sensitized OLEDs with narrow-band emission were developed. And the effect of conformation distribution was also investigated in the EL process. We believe that no overlapped contexts and conclusions are shared in our work with the existing literatures. The statements in introduction were modified to better demonstrate the novelty.

2. I cannot understand the reason why the authors stated “a very large k_{RISC} of $2.76 \times 10^6 \text{ s}^{-1}$ can be calculated for TBP-3MCz.” Most of TADF emitters show k_{RISC} values of $> 1 \times 10^6 \text{ s}^{-1}$. Moreover, to date, several excellent TADF emitters were reported to show a k_{RISC} as large as $> 10^7 \text{ s}^{-1}$ (*Nat. Photonics* 14, 636–642 (2020); *Nat. Photonics* 14, 643–649 (2020)).

Response: We understand that the k_{RISC} of some TADF materials were reported to be $> 10^7 \text{ s}^{-1}$, and we have referred to the literatures reporting high RISC values ($k_{\text{RISC}} > 10^6$) in the second paragraph in introduction. We change the statement to “a large k_{RISC} ”. However, “most of TADF emitters show

k_{RISC} values of $> 1 \times 10^6 \text{ s}^{-1}$ is invalid. Only a few TADF materials can achieve fast spin-flipping among the reports on TADF materials developed in decades (k_{RISC} typically range from 10^3 to 10^5 s^{-1}), and most of them have been cited in the introduction. So far, the efficiency roll-off issue is still one of the obstacles for TADF-OLED because of the long-lived triplet excitons. For this reason, developing TADF materials with short excited state lifetime as emitters and sensitizers is one of the most important topics in this field, in which structure-property relationships are needed. In our manuscript, we discussed the effect of conformation distribution on the RISC process and provided molecular design strategy to shorten the excited state lifetime. The molecules in our work with confined conformation distribution can achieve $k_{\text{RISC}} > 10^6$, short excited state lifetime and low efficiency roll-off in OLED. We believe that our work can extend the scope for TADF material development.

In addition, the k_{RISC} values can be varied depending on the fitting protocols, measurement conditions (solution or film state) and the adopted kinetic models. The k_{RISC} of the 15 wt% TBP-3MCz: PhCzBCz film varied from 2.76×10^6 to $3.69 \times 10^6 \text{ s}^{-1}$ using different kinetic models, and TBP-3MCz in toluene solution can have a large k_{RISC} of $1.3 \times 10^7 \text{ s}^{-1}$. Moreover, if the materials possess very short prompt lifetime and large delayed percentage, the calculated k_{RISC} would be very large, while the delayed lifetime is still in microsecond time-range. Therefore, the k_{RISC} value is difficult to compare or benchmark in different works. Nevertheless, developing materials with both high efficiency and short excited state lifetime is important for OLED industry and we believe our work can be helpful to achieve this goal.

3. I would suggest that the authors used range-tuned ωB97XD^* functional in both geometrical optimization and TDDFT calculations.

Response: Thanks for your suggestion. B3LYP functional is the most general functional in TADF research and in combined with DFT-D3 correction, it can accurately optimize the molecular geometries with reduced cost (*J. Chem. Theory Comput.* 2015, 11(4), 1481–1492). For the excited state, we used range-tuned ωB97XD^* functional for TD-DFT calculation. However, the range separate parameters in ωB97XD^* are optimized based on the fixed pre-optimized geometry. So, it is difficult to use it for optimization. For safety, we compare the optimized S_0 geometries using B97XD and B3LYP(D3) functionals and TD-DFT calculated energies (based on the optimized S_0 geometries in the same $\omega\text{B97XD}^*/\text{def2SVP}$ level), which show negligible differences.

Supplementary Figure 5. Comparison of the optimized geometries using different methods and the vertical transition energies based on the optimized geometries (in $\omega\text{B97XD}^*/\text{def2-SVP}$ level).

4. The authors claimed that the conformation-dependent SOCME of TBP-MCz is small, but the variation is significant (Fig. S5). Moreover, the higher triplet states, especially the LE triplet states might play important roles as conformation varies.

Response: Thanks for your questions. The LE state of TADF materials with large SOC effect plays an important role in the spin-flipping process, especially for the blue TADF emitter with close CT energy to LE state, and this has been proved by many literatures. However, in our case, the donor and acceptor fragments have high energy according to their phosphorescence spectra (>3.0 eV, Supplementary Figure 8), the large energy gap decreases the possibility for interactions between the CT state and LE state. In theoretical simulation, the $n-\pi^*$ transition triplet state of the TBP moiety (T_7) with large SOC is 0.4 eV higher than the singlet state of TBP-3MCz, which indicates that it would have little interactions with the S_1 state. And the T_4 state with closer energy is a long-range CT state with small SOC. Moreover, the high energy T_2 state is >0.2 eV above the S_1 state among the possible conformers of TBP-1MCz (40° - 140°) on the potential surface, therefore, the influence of the higher triplet state is less significant. As the LE and CT state energy relationship can be altered by environment polarity, we measured the transient PL decay characteristics of the emitter in hosts with different polarity (Supplementary Figure 9). They show insignificant changes, which indicates that the LE state is, at least, not the most contributing factor in spin-flipping. Combining an LE state with large SOC with this conformation distribution strategy might be helpful to further accelerate the RISC process, and can be a research topic in the future.

Figures from, Supplementary Figure 9 and Supplementary Figure 10:

Supplementary Figure 6. Excited state energy levels and hole-electron distributions of the high-lying triplet states of TBP-3MCz. **Supplementary Figure 10.** PES of the S_1 , T_1 and T_2 states of TBP-1MCz. **Supplementary Figure 9.** d) Transient PL decay characteristics of 15 wt% TBP-3MCz in hosts with different polarities.

The “small” SOC values we mentioned in the manuscript is to compare the SOC values with the reported TADF emitters incorporating heavy atoms such as sulfur and selenium (*Angew. Chem. Int. Ed.* 2022, e202205684; *Nat. Photon.* 2022, 16, 803–810), and our investigated molecules do not involve the heavy atom effect. The statement was modified in the manuscript to avoid ambiguity. Inspired by the question, we tried to obtain a conformation-SOC-RISC relationship using Fermi’s golden rule equation $k_{RISC} = \frac{2\pi}{\hbar} H_{SO}^2 (4\pi\lambda k_B T)^{-1/2} \exp\left(\frac{-\Delta E_{ST}}{k_B T}\right)$. However, the obtained rate constants show three orders of magnitude deviations, and they are highly sensitive to the value of H_{SO} , ΔE_{ST} and λ . Moreover, the SOC calculated by PySOC code with an effective charge

approximation is not accurate enough (*J. Chem. Theory Comput.* 2017, 13, 515). Conducting quantitative investigation needs high level calculations for accuracy such as the work in *J. Am. Chem. Soc.* 2017, 139, 4042, which is beyond the scope of this manuscript.

5.Line 122, delta EST should be obtained from Supplementary Fig. 2a. Moreover, hole-electron distributions of the triplet states of TBP-3aDMAC should be provided to support their claim of LE character (Line 120).

Response: We are sorry for the typo. They have been corrected. We calculated the hole-electron distributions of TBP-3aDMAC. Considering that an aDMAC donor would have quasi-axial (QA) or quasi-equatorial (QE) conformations, there would be four possible conformers in one molecule: all QA, two QA and one QE, one QA and two QE, and all QE. The hole-electron distributions were depicted in Supplementary Figure 7 and the energies were summarized in Supplementary Table 3. From the TD-DFT calculation results, if QA and QE conformers coexist in one molecule, the lowest singlet state is originated from the twist QE moiety and the lowest triplet state is originated from the planar QA moiety with large hole and electron overlap. This is in good accordance with the CT-type singlet state and LE-type triplet state of TBP-3aDMAC in spectral characterizations. In combined with the experimental results, in the film state, the high energy QA state would undergo internal conversion to the QA lowest singlet state or fast Förster energy transfer to another molecule with QE conformer. Therefore, fluorescence from low energy QE conformer is dominated in the 15 wt% doped film. However, the triplet state distributes on the QA conformer. After ISC or generation of triplet excitons in the EL process, the QA conformer distribution with large ΔE_{ST} would prolong the excited state lifetime. Multiple conformation distribution is the extreme case of conformation distribution, which again highlight the importance of conformation distribution management for short lifetime TADF materials. These discussions were added in the main text to reveal the multiple conformation characteristics of TBP-3aDMAC.

Supplementary Figure 7. Optimized geometries of TBP-3aDMAc and the hole-electron distributions of singlet and triplet states.

6. Single crystal results are necessary to support their claims.

Response: Thanks for your suggestions. It is difficult to obtain single crystals possibly due to their large molecular weight. We can only obtain the single crystal of TBP-3aDMAc (CCDC: 2191190). It shows a quasi-axial conformation with small dihedral angle and a staggered arrangement in the single crystal structure. The single crystal shows a blue emission, and the PL spectrum peaks at around 430 nm. This indicates that the quasi-axial conformer possesses a higher energy emission, which is in accordance with the theoretical calculation results.

Supplementary Figure 4. Single crystal structure of TBP-3aDMAc with quasi-axial conformation (CCDC: 2191190) and its PL spectrum.

7. For TBP-a-DMAC, conformations should be much more complicated than the authors discussed. There should be at least 2*2*2 types of conformers.

Response: Thank you for your comments. Indeed, all QA and all QE conformation is just extreme cases. As discussed in the response of question 5, in single molecule, there can be four local minimums of conformation. But some conclusions can be drawn based on the simplification such as the CT-type singlet and LE-type triplet state. In reality, the dihedral is dynamically changing. Therefore, we used MD simulation and counting the conformation and energy distributions from multiple frames to simulate this complex situation. As shown in Fig. 3, there are QA conformer distributions with small D-A dihedral angle. And the TD-DFT calculated energy distribution based on random 50 geometries have taken the dynamic conformation distribution into consideration, which result in the wide distribution of singlet and triplet state. This wide distribution was also proved by the time-resolved PL and EL spectra and long delayed lifetime.

8. What are the distributions of QA and QE conformers in amorphous states? More solid experimental results should be provided rather than only theoretical estimations.

Response: It is difficult to estimate the specific percentages of QA and QE conformers. One way to calculate the percentage of QA and QE conformers is based on conformation energies obtained from DFT and TD-DFT calculation and Boltzmann distributions. This method was adopted by many literatures such as *Angew. Chem. Int. Ed.* 2021, 60, 25878 and *Chem. Sci.* 2019, 10, 10687. Using this method, the percentage would be different in ground state, singlet state and triplet state, which is highly dependent on the shape of the potential surface. We calculated the percentages of different conformers in different state in Supplementary Table 4. Another possible method is to separate the absorption from QA and QE conformers and calculate based on the oscillator strengths of each conformer. However, the situation is very complex in our system having multiple donors, and the absorption spectra is difficult to separate. Therefore, we can only simulate the conformation distribution by molecular dynamic methods. Nevertheless, the main point for our manuscript is to find a structure-property relationship between conformation distribution and excited state lifetime of TADF materials. Dual conformation is a typical and extreme situation for broad conformation distribution, and our work does not devote to analyze the percentage of conformers quantitatively. With the additional experiments including time-resolved PL and EL spectra in Figure 4 and the discussions in the next question, we believe that the broad triplet energy distribution effect of the TBP-3aDMAC with conformation heterogeneity can be revealed without precise percentage of QA and QE conformers.

Supplementary Table 4. TD-DFT calculated ΔE_{ST} in the ground state geometries (QA or QE conformer) and the average ΔE_{ST} according to Boltzmann distribution on the scanned S_1 and T_1 PES in Fig 2.

	ΔE_{ST} in S_0 geometry meV	S_1 average ΔE_{ST} ^{a)} meV	T_1 average ΔE_{ST} ^{a)} meV
TBP-1MCz	40.4	23.6	25.6
TBP-1DMAC 462 (QA 33%)/7.7 (QE 67%) ^{b)}		25.4	75.4
TBP-1aDMAC 463 (QA 94%)/8.5 (QE 6%) ^{b)}		28.7	153.8 (QA 11%/QE 89%)

a) The conformer distributions were estimated by Boltzmann distribution: $\%Conformer i =$

$$\frac{\exp\left(-\frac{E_i}{k_b T}\right)}{\sum_j \exp\left(-\frac{E_j}{k_b T}\right)},$$

where E_i is the conformational energy of conformer i calculated by DFT or TD-DFT,

k_b is Boltzmann constant and T is the ambient temperature (298 K); and the average energies were calculated by timing TD-DFT calculated ΔE_{ST} in each geometry with the $\%Conformer i$ of the corresponding scanned singlet or triplet PES.

b) The ΔE_{ST} calculated in quasi-axial (QA) and quasi-equatorial (QE) conformation and the percentages of each conformation calculated by the geometry energy and Boltzmann distribution.

9. In Fig. S6, the authors simply ascribe the variation of time-resolved PL spectra to the solvent effect and conformation heterogeneity. Possible energy transfer between different conformers should be considered. Should the long-tail conformer show a fast energy transfer to the twisted conformer or not? The authors should explain.

Response: We appreciate the question. Energy transfer between conformers including singlet and triplet energy transfer is an important factor to be considered. Comparing the PL spectra of TBP-3aDMAc in PMMA, the short-wavelength emission shoulder is not conspicuous, and can only be observed when changing the excitation wavelength at 1 wt% doping concentration. Moreover, the transient fluorescence decays of the shoulder of the films with different doping concentrations were measured. The lifetime at short time integral decreases slightly when increasing the doping concentration, indicating the efficient Förster energy transfer (FRET) between the conformers, possibly due the coexistence of dual conformer in single molecule as discussed in question 5. The FRET is efficient even at 2 wt% doping concentration. And the high energy conformer of the 15 wt% TBP-3aDMAc: PhCzBCz film can only be observed using time-resolved emission spectra at very short time integral. Therefore, the fluorescence from the QE conformer is dominated in 15 wt% film state due to the fast energy transfer.

Supplementary Figure 3. c) concentration-dependent PL spectra of TBP-3aDMAc in PMMA host; b) excitation and PL spectra of the 1 wt% TBP-3aDMAc: PMMA film; f) Transient PL decays of

the TBP-3aDMAc: PhCzBCz film with different doping concentrations measured at 450 nm for the quasi-axial emission band; g) Time-resolved PL spectra of the 15 wt% TBP-3aDMAc PhCzBCz film in ns time scale.

However, as the triplet potential surface is flat as shown in Figure 2d, the triplet energy distribution should be more conspicuous in the host-guest system. Moreover, due to the short-range exchange mechanism of the triplet exciton migration, Dexter energy transfer cannot be so efficient in the host-guest system. The triplet energy distribution from a broad conformation distribution would have a significant influence on the delayed fluorescence, as revealed by the obvious spectral shift of TBP-3aDMAc in the time-resolved PL and EL spectra in Figure 4. To further confirm the triplet energy distribution effect, we compare the time-resolved PL of 15 wt%TBP-3aDMAc in CBP host. The selection of the CBP host is because it has similar molecular structure and polarity as the PhCzBCz host, but very closed triple energy to the TBP-3aDMAc, which allows for the diffusion of triplet excitons. The 15 wt%TBP-3aDMAc: CBP film shows a double exponential decay with shorter lifetime in compared with the 15 wt%TBP-3aDMAc: PhCzBCz film. Similarly, the prompt emission (integrated from 0-500 nm) possesses the short wavelength portion emission while the delayed emission shows a red-shift and narrower spectrum. This indicates the prompt emission is partly originated from the high energy QA conformer and the delayed emission is dominated by the QE conformer. Notably, the time-resolved PL of the delayed component shows a negligible change as delayed time, which is very similar to that of the 15 wt% TBP-3MCz: PhCzBCz film (Figure 4). Considering the exciton diffusion and quenching process in the low triplet host CBP, this can be explained by the triplet excitons in the partial conformer distributions with large ΔE_{ST} were quenched or diffused to the partial conformer distributions with small ΔE_{ST} , as indicated by the scheme. This can further confirm the triplet energy distribution effect of the TBP-3aDMAc with conformation heterogeneity, which it the key factor for the long TADF lifetime in OLED. The discussion and data were supplemented to the manuscript and Supplementary Figure 12.

Supplementary Figure 12. Investigation of the triplet state distribution and exciton diffusion in the host-guest system. a) Phosphorescence spectra of CBP neat film, 15 wt% TBP-3aDMAc: CBP film

and 15 wt% TBP-3aDMAc: PhCzBCz film; b) Comparison of the steady state PL spectra, prompt fluorescence spectra (0-500 ns) and delayed fluorescence spectra ($> 1 \mu\text{s}$) of the 15 wt% TBP-3aDMAc: CBP film; c) comparison of the transient PL decay of the 15 wt% TBP-3aDMAc doped CBP and PhCzBCz films; d) time-resolved PL spectra of the delayed components of the 15 wt% TBP-3aDMAc: CBP film; e) schematic illustration of the TADF emission of TBP-3aDMAc with different conformers in host with high or low triplet energy.

10. Have the authors tried to fabricate evaporation-processed OLEDs? Based on the TGA results, I think preparing evaporation-processed OLEDs might be also available.

Response: Thanks for your questions. We fabricated the evaporation-processed OLEDs using the same EML with a device structure of ITO/ HATCN (5 nm)/ TAPC(30 nm)/ TCTA (10 nm)/ PhCzBCz (10 nm)/ 15 wt% TADF emitter: PhCzBCz: (30 nm)/ PPF (10)/ TmPyPb (40 nm)/ LiF (1 nm)/ Al. The OLEDs based on TRZ-3MCz and TB-3MCz show a maximum EQE of 28.2% and 30.1%, respectively, along with low efficiency roll-off. The improved device efficiency is mainly due to the horizontal dipole orientation in the vacuum-evaporated film (horizontal dipole ratio of 89%) that increase the light out-coupling, as revealed by the angular-dependent of the emission intensity of the *p*-polarized light. The results were added in Supplementary Figure 20. However, TBP-3MCz only shows a 15% EQE in evaporation-processed OLED. We suppose that the benzophenone moiety might be unstable in the thermal evaporation process, as shown by the plate TLC results of the TBP-3MCz material from the quartz boat after device fabrication.

Supplementary Figure 20. Characterization of the vacuum-evaporated OLED devices. a) *J-V-L* curves and b) EQE-luminance curves (insert: EL spectra of the devices driven at the current density of 1 mA cm⁻²) of the vacuum-evaporated TADF OLEDs; angular-dependent of *p*-polarized PL intensities of c) 15 wt% TRZ-3MCz: PhCzBCz and d) 15 wt% TB-3MCz: PhCzBCz films. **Figure R1.** Plate TLC results of the TBP-3MCz material from the quartz boat after device fabrication.

11. Device operation lifetime should be measured as it is an important claim in the abstract.

Response: Thanks for your suggestion. The operation lifetimes of the abovementioned OLED devices were measured. The T_{50} of the bluish-green emitter TRZ-3MCz is 212 hours and that for the sky-blue emitter TB-3MCz is 16 hours, at the initial luminance around 1000 cd m^{-2} . The operation lifetime of DMAc-TRZ based on the same device structure was also measured for comparison. The device lifetime could be further improved by replacing the less stable carrier transport materials such as TAPC and TmPyPb and device optimizations.

Supplementary Figure 20. e) Luminance of the OLED devices as a function of operation time at initial luminance of around 1000 cd m^{-2} .

Reviewer #2 (Remarks to the Author):

Three new TADF emitters TBP-3MCz, TRZ-3MCz and TB-3MCz have been synthesized, with emphasis on multiple rigid donor components to confine the conformation of the emitters. Highly-twisted geometries give small ST gaps. Fast RISC rates are demonstrated leading to very efficient solution processed OLEDs, with low efficiency roll-off at high brightness. These are currently important topics in the field. This is noteworthy. The article is logically written. The molecular design is rationally presented. Standard photophysical, structural characterization techniques and device fabrication are used, although the photophysics is rather limited.

The manuscript could potentially become suitable for Nature Comm. But major revision will be required as listed below.

Response: Thanks for the positive evaluation of our work. The additional photophysical characterizations were added to strengthen our results, especially for the dual-conformation phenomenon.

1. Established literature: Conformational control is important in donors and there is discussion in the article of axial, equatorial acridines. Quasi-axial conformer is stated to be TADF inactive. However, this is not a new concept. Established literature is not referenced. Acridine conformers have also been discussed previously, Li, W et al, *Angew Chem Int Ed Engl.* 2019, 58(2):582-586. doi: 10.1002/anie.201811703. This is also well established with phenothiazine donor in D-A molecules: Chen, C et al *Angew. Chem. Int. Ed.* 2018, DOI: 10.1002/anie.201809945.

Response: Thanks for your comments. The dual conformation phenomenon has been investigated widely, and the dual conformation emitters possess abundant special photo-physical characteristics such as multiple emissions and room temperature phosphorescence. However, the previous investigation on dual conformation pays less attention on the excited state lifetime and RISC process,

which is important for efficient TADF OLED. The mentioned adamantane-substituted acridine donor was developed by our group (Li, W et al, *Angew. Chem. Int. Ed.* 2019, 58, 582), in which we achieve blue dual emission and high efficiency TADF OLED. However, the OLED efficient roll-off was unsatisfactory. The previous research is one of the research backgrounds of this work and we take the aDMAc donor with dual conformation as a typical and extreme example for conformation distribution. In this work, we devoted to investigate the role of conformation distribution on the key RISC process for TADF materials, and dual conformation is the case for wide conformation distribution. The wide conformation is not favorable for spin-flipping, so we propose a conformation distribution control strategy by rigid donors, which was further proved by the three prototype TADF emitters. The concerning dual conformation literatures were listed and discussed in the introduction section.

2. Data analysis. It is stated in SI “Energy transfer from high energy quasi-axial to low energy quasi-equatorial form can happen.” This energy transfer between the two conformers should be analyzed in more detail experimentally. Does this depend on the solid-state host? Only PMMA host is reported (manuscript page 4, line 1; SI Fig 2).

Response: We are grateful for the question. We measured the concentration-dependent PL spectra of TBP-3aDMAc in PhCzBCz host in addition to the PMMA host, which show similar trends. According to the TD-DFT calculations, PL spectra in DCM and PMMA host (1 wt%), and PL spectra of TBP-3aDMAc single crystal with QA conformation, we can find that the QA conformer possesses high energy emission at around 450 nm in film state. The FRET from QA to QE conformer can be revealed by transient PL measurement on the 450 nm band. As doping concentration increases, the prompt emission decays more quickly, indicating the efficient FRET process in the film state. Moreover, due to the possible coexistence of QA and QE moiety in one molecule, the internal conversion of the high-lying excited state from QA to the lowest excited state from QE moiety would be very fast (which is also discussed by TD-DFT calculation, please refer to the answer of the question 5 from Reviewer#1). Therefore, at high doping concentration, the FRET and internal conversion of the QA conformer is fast and the emission is dominated by the low energy QE conformer. In the 15 wt% TBP-3aDMAc: PhCzBCz film, the high energy emission can only be observed by time-resolved PL spectra in 1 ns time-scale. However, despite the efficient energy transfer of singlet state, the QA conformer distribution with large ΔE_{ST} would have great influence on the triplet exciton dynamics, resulting in long TADF lifetime. Moreover, the emission of TBP-3aDMAc also shows a normal bathochromic shift in hosts with different polarity, which is the same as the most TADF materials.

Supplementary Figure 3. Concentration-dependent PL spectra of TBP-3aDMAc in c) PMMA and d) PhCzBCz host (excited at 320 nm); b) excitation and PL spectra of the 1 wt% TBP-3aDMAc: PMMA film; f) Transient PL decays of the TBP-3aDMAc: PhCzBCz film with different doping concentrations measured at 450 nm for the quasi-axial emission band; g) Time-resolved PL spectra of the 15 wt% TBP-3aDMAc PhCzBCz film in ns time scale. **Supplementary Figure 9.** Transient PL decay characteristics of 15 wt% TBP-3aDMAc in hosts with different polarities.

3. Data analysis. Emitter layer on OLEDs is 15% doped in PhCzBCz. There should be more discussion on the choice of host and why 15% was chosen. I cannot see data about optimization of dopant concentration.

Response: Thanks for your question. We measured the basic photo-physical properties of TBP-3MCz in PhCzBCz host with different doping concentrations (Supplementary Figure 1e and 1f). As the doping concentration increases, the spectrum redshifts while the prompt and delayed fluorescence lifetimes decrease. This is a normal phenomenon in TADF because of the increased polarity and exciton quenching from intermolecular interactions. Similar phenomenon can also be found in TBP-3aDMAc. We had conducted device optimization by varying doping concentration and found that the 15% doping concentration in PhCzBCz host can achieve the highest EQE and low efficiency roll-off. The 5 wt% device has lower efficiency and luminance can be attributed to insufficient energy transfer while further increasing the doping concentration it would suffer from poor carrier transportation and exciton quenching. The other emitters also show the best performance at 15 wt% doping concentration. Moreover, at high doping concentration, the involvement of Dexter energy transfer and exciton quenching would increase the complexity for our investigation on conformation distributions. Therefore, we chose 15 wt% concentration for investigation.

Supplementary Figure 1. e) PL and f) transient PL decay characteristics of the TBP-3MCz: PhCzBCz films with different doping concentrations.

Figure R2. Solution-processed OLED device performance based on TBP-3MCz with different doping concentrations.

4. Neat emitter layer, (non-doped) devices could also be reported for comparison.

Response: Thanks for your suggestion. We fabricated non-doped solution-processed OLED with a device structure of ITO/PEDOT: PSS (40 nm)/ non-doped EML (30 nm)/TmPyPb (40 nm)/CsF (1 nm)/Al for comparison. Unfortunately, both the luminance and efficiency are very low, with the highest EQE less than 2.5%. As shown in the answer of question 3, the non-doped film shows both significantly reduced prompt and delayed lifetimes. This is mainly due to the aggregation caused quenching effect of fluorescence and triplet-triplet annihilation of triplet excitons as these molecules have no shielding groups to prevent the intermolecular interactions.

Figure R3. Solution-processed OLED device performances based on non-doped emission layers.

5. SI. For further unambiguous proof of molecular structures, copies of all mass spec should also be shown as figures in SI.

Response: Thanks for the suggestions. They have been attached to SI.

6. Methodology: Quantum yields. Table 1, Supplementary Table 4 and elsewhere. It is not meaningful to quote photoluminescence quantum yields to one- or two- decimal places level of accuracy, as reported here. This suggests that the authors do not appreciate the inherent inaccuracy in the technique (usually +/- 10% of the value reported). For example, Table 1, line 1, 79.6, 20.59 and 59.01 should be 80, 21 and 59.

Response: Thanks for your suggestions. We also agree that the PLQY cannot be so accurate. The PLQY values in the manuscript were modified. Also, the photo-physical parameters were recalculated based on the PLQYs with integral number, while the change in decimal has less influence on the rate constants.

7. Methodology: Supplementary Table 5. EQEs cannot be accurate to two decimal places. Please correct this.

Response: Thanks for your reminder. They have been corrected.

Reviewer #3 (Remarks to the Author):

This work examined the influence of the distribution of dihedral angles between donor and acceptor moieties within a TADF emitter on the photophysical properties, such as lifetime, emission profiles. Molecular Dynamic simulations have been used to model the distribution of emitter inside the matrix of host. It was found that emitters that are structurally more rigid showed a narrower conformational distribution, while one emitter showed 2 dominant conformations gave rise to unfavorable longer emission lifetime. However, such effect on TADF behavior is well known in the

literature (e.g. J. Phys. Chem. A 2021, 125, 35, 7644)

Response: Thanks for the comments and mentioning some key points of our work. The conformation of organic emitters has significant influence on the photo-physical properties including dual emission and room-temperature phosphorescence and therefore, the structure-property relationships have been investigated heavily before in different aspects. Some key literatures have been cited in the introduction. On the theoretical basis, such as the listed literature by the reviewer and *Angew. Chem. Int. Ed.* 2022, 61, e2022134, the researches devoted to find an optimal conformation for efficient TADF, which are on single molecule basis. However, as mention in the introduction, the molecules in emission layer and OLED are in amorphous state, where the conformation cannot be fixed. In our study, viewing from the conformation distribution in host-guest system, we investigate the influence of conformation distribution in host-guest system on the key RISC process of TADF materials and put forward a molecular design strategy to achieve fast spin-flipping, which was further verified by OLED device characterization. Moreover, our investigation involves molecular simulation, photophysical investigation and device characterization, hoping to recognize the effect of molecular conformation distribution on the excited state lifetime and OLED performance from theoretical to practical. Therefore, we believe that our work provides a new scope on excited state lifetime of TADF materials and shade light on developing OLED with high efficiency and low efficiency roll-off.

1. The use of the term “long-tail” seems rather unscientific and sometimes unclear. Would there be other way to describe it?

Response: Thanks for the suggestions. A long tail distribution is a distribution that has a long “tail” that slowly tapers off toward the end of the distribution, which is always ignored but has important influence. We hope to refer to the partial conformers with large ΔE_{ST} in film state that prolongs the excited state lifetime of TADF materials. It seems that it is a little exaggerated. We change the statement to “partial conformer distribution with large singlet-triplet energy gap” to be more specific.

2. From line 123 onwards, the authors tried to demonstrate the dual conformational character of TBP-3aDMAc by concentration-dependent PL spectra in PMMA and excitation dependent solution emission spectra. According to the authors, increasing doping concentration led to red-shift of emission due to more efficient energy transfer from the quasi-axial to quasi-equatorial conformers. But why would the emission band of the neat film occur at a higher energy than that at 50wt% doping concentration? Could the red shift of emission band be also explained by the increasing polarity of the film with increasing doping concentration?

Response: We appreciate your question. We measure the concentration-dependent PL spectra of the TBP-3aDMAc: PMMA film again. It shows a normal red-shift as doping concentration increases, which is due to the increased polarity of the film. And similar tendency is also found in the TBP-3aDMAc: PhCzBCz films. The previous spectra were mixed by the data from different spectrometers (FluoroMax-4 and FL980), which will lead to the slightly different spectral shapes even for the same film sample. Now all the data were obtained from the same FL980 spectrometer to avoid ambiguity. We are very sorry for the error.

Supplementary Figure 3. Concentration-dependent PL spectra of TBP-3aDMAc in c) PMMA and d) PhCzBCz host (excited at 320 nm).

In the solution state, the molecules should have more degree of freedom for bond rotations and therefore would be easier to access other conformations and therefore assignment of only two conformation based on solution state study may not be concrete enough. Due to the highly twisted nature of these molecules, electronic communications between different parts of the molecule may not be efficient resulting in dual emission from just a single conformer. Did the author measure any excitation spectra to exclude such possibility?

Response: As shown in the answer of question 5 from Reviewer#1, there might be four possibilities in one molecule: all QA, two QA and one QE, one QA and two QE, and all QE. From the TD-DFT calculation results, if QA and QE conformers coexist in the molecule, the lowest singlet state is originated from the twist QE conformer and the lowest triplet state is originated from the planar QA moiety with large hole and electron overlap. This is in good accordance with the CT-type singlet state and LE-type triplet state of TBP-3aDMAc in the spectral measurements. We add the excitation scans of the two bands in dichloromethane solution to better illustrate the dual emission character, which shows a difference at short wavelength region. Similar phenomenon can also be found in the 1wt% TBP-3aDMAc: PMMA film. It is really hard to tell the short wavelength emission is from the molecule with all QA conformer or anti-Kasha emission from the high energy QA conformer (because of the large oscillator strength of the planar conformer) of one molecule because we cannot “see” the emission from single molecule. Nevertheless, in the film state, the fluorescence energy transfer or internal conversion from QA to QE conformer should be very efficient as shown in the answer of question 2 from Reviewer2#. And in the 15 wt% doped film and OLED devices where we mainly focus on, the CT fluorescence comes from the QE conformer and the triplet state would be influence by the hidden QA conformer distribution.

Supplementary Figure 7. Optimized geometries of TBP-3aDMAc and the hole-electron distributions of singlet and triplet states. **Supplementary Figure 3.** a) UV-vis absorption, excitation and PL spectra of the TBP-3aDMAc dichloromethane solution (10^{-5} M).

The vibronic feature in the emission spectra also shows resemblance with the phosphorescent spectra of just the donor part aDMAc. The author may also compute the emission spectra based on the two conformers that have been optimized.

Response: We compared the PL spectra of TBP-3aDMAc in DCM solution with the phosphorescence spectrum of aDMAc and the fluorescence spectrum of TBP-3aDMAc single crystal with QA conformation. It is difficult to obtain phosphorescence emission especially solution state where the non-radiative transition is significant, and phosphorescence also cannot be observed in the PL spectrum of aDMAc solution at room temperature. As advised by Reviewer#1, we measured the fluorescence spectrum of single crystal of TBP-3aDMAc with QA conformation, which is close to the high energy emission in DCM (also influenced by solvent polarity). In combined with the TD-DFT calculated energy of the QA state, it is safe to attribute the emission at short wavelength to the emission from the QA conformation with LE character. Due to the large geometry change in the excited state of the D-A molecule, vibrational-resolved emission spectra cannot be calculated by Gaussian (the Franck-Condon factor corresponding to the overlap integral between both vibrational ground states is too small). But the origin of the two emissions can be proved by the calculated excited state energies and additional experimental results.

Figure R4. Comparison of the PL spectrum of TBP-3aDMAc in DCM solution (excited at 320 nm), phosphorescence spectrum of aDMAc donor and PL spectrum of TBP-3aDMAc single crystal.

3. Lines 158-160, “Locally excited states of the donor and acceptor moieties...can hardly interact with the low-lying CT triplet state...”. Try to tone down a bit.

Response: Thanks for your suggestion. The LE state of TADF materials with large SOC effect plays an important role in the spin-flipping process, especially for the blue TADF emitter with close CT energy to LE state, and this has been proved by many literatures. However, in our case, the donor and acceptor fragments have high energy (>3.0 eV, Supplementary Figure 4). In theoretical simulation, the $n-\pi^*$ transition triplet state of the TBP moiety with large SOC (T_7) are 0.4 eV higher than the singlet state, which indicates that it would have little interactions with the S_1 state. As the LE and CT state energy relationship can be altered by polarity, we measured the transient PL decay characters of the emitter in hosts with different polarity. They show insignificant changes, which indicates that the LE state is, at least, not the most contributing factor in spin-flipping. The statement in the main text was modified. Combining an LE state with large SOC with this conformation distribution strategy might be helpful to further accelerate the RISC process, and can be a research topic in the future.

Supplementary Figure 6. Excited state energy levels and hole-electron distributions of the high-lying triplet states of TBP-3MCz. **Supplementary Figure 9.** d) Transient PL decay characteristics of 15 wt% TBP-3MCz in hosts with different polarities.

4. Lines 164-165, “For simplification, only one donor connected to the TBP acceptor was allowed to motion move in the scanning”, does it mean there are two other donor moieties on the other part

of the molecule that are fixed, but I don't see those in your structure in Fig 2. The authors need to clarify that. If there is only one donor is used in your calculation (to reduce computational cost), can it truly reflect the conformational distribution as in the molecules you synthesized?

Response: As shown in Fig. 2a, the structures for potential surface scanning possess only one donor (MCz, DMAc or aDMAc) connected to the TBP acceptor. We make this simplification because the multiple donor structure possesses multiple degenerate states, which would increase the difficulty to analyze the results. And if we fix one donor, the lowest singlet state would be distributed on the one with the largest dihedral angles (lowest CT state energy), just like the case of TBP-3aDMAc with the coexistence of QA and QE conformers. Moreover, the multiple donor structure also increases the computational cost. To illustrate the representativeness, we depicted the hole-electron distributions of TBP-1MCz (calculated in the same level as the PES scanning) and compared it with that of TBP-3MCz. The hole mainly distributes on the MCz moiety and the electron mainly distributes on the benzophenone moiety, resulting in the similar small ΔE_{ST} in the optimized geometry. Therefore, TBP-1MCz can be representative for the conformation relationship of TBP-3MCz. In the main text, we change our statement to be clearer: "For simplification, the scanned structure process only one donor connected to the TBP acceptor, which can be representative because it has similar excited state character and energies with the degenerated states of the structure with multiple donors". In the MD simulations and TD-DFT calculations in Fig. 3, the guests are the molecules with multiple donors, which can reflect the conformational distributions of the host-guest systems.

Supplementary Figure 10 and Supplementary Figure 6. Molecular structure of TBP-1MCz; hole-electron distributions of singlet and triplet state of TBP-1MCz and TBP-3MCz.

5. Line 245, What does it mean by solid state solvent effect stabilization? Solvent should be gone at this point, in theory.

Response: In the host-guest film state, the guest is surrounded by the host molecules and the hosts act like the "solvent". In solution state, the large transition dipole moment of the excited state molecule would induce strong dipole-dipole interactions with solutions and the solutions reoriented rapidly (in picoseconds). This effect is the reason for bathochromic shift of CT emitters in solvent with large polarity. In the film state, the dipole rearrangement of the host is slow, usually in ns time-scale. Therefore, in the time-resolved PL spectra, the spectral red-shift as delayed time can be overserved in the first a few ns. (Detailed investigations can be found in *Adv. Optical Mater.* 2019, 7, 1801644; *Nat. Mater.* 2022, 21, 1150 and *J. Mater. Chem. C* 2017, 5, 11001) We also make some

explanations in the manuscript.

6. Line 251, It would be better to also show the TR-spectra of TBP-3MCz in thin film state for comparison.

Response: Thanks for your question. The previous version of the manuscript has shown the time-resolved PL spectra in Figure S6. Considering that they are important evidence for conformation heterogeneity, we move it to the main text (Figure 4) and make some modification for better comparison. In addition, we also measured the time-resolved electroluminescence spectra to investigate this effect in OLED device. In both PL and EL conditions, the delayed spectra of TBP-3aDMAc show the most conspicuous changes as delayed time, while the TBP-3MCz with confined donor conformation has negligible changes. The spectral changes of TBP-DMAc and TBP-3aDMAc is due to the different TADF lifetimes of different conformers. This distinct difference in both PL and EL conditions clearly shows that the confined conformation distribution of TBP-3MCz, which is the reason for short excited state lifetime. Detailed discussion can be found in the revised manuscript.

Fig. 4. Time-resolved spectra investigations of the doped TBP-3MCz, TBP-DMAc and TBP-3aDMAc host-guest systems. **a** Comparison of the steady state PL spectra, prompt fluorescence spectra (500 ns) and delayed fluorescence spectra ($> 1 \mu\text{s}$); **b** time-resolved PL spectra of the delayed components after solid-state solvent stabilization and fluorescence emission; **c** time-resolved EL spectra of the OLED devices based on the host-guest systems (measured with pulse voltage of 6 V and duration of 300 ns).

7. Fig 2c and d. Could the author explain why only for the case of TBP-a-DMAc, there is some distribution at angles around 0° , but it is not seen in TBP-DMAc even though they showed very similar energy trends for ground state, singlet and triplet excited states?

Response: Thanks for your question. As shown in the caption of Supplementary Table 4, the conformation distributions were calculated by Boltzmann distribution: $\%Conformer i =$

$$\frac{\exp\left(-\frac{E_i}{k_b T}\right)}{\sum_j \exp\left(-\frac{E_j}{k_b T}\right)},$$

where E_i is the conformational energy of conformer i calculated by TD-DFT, k_b is

Boltzmann constant and T is the ambient temperature (298 K). Therefore, the distribution depends on the relative energy relationships. For the ground state potential surface of TBP-1DMAc, there are two energetically close local minimum, and therefore, there will be conformation distribution on both 0° (QA) and 90° (QE), with ΔE_{ST} of 462 and 7.7 meV, respectively. Similar potential energy surface shapes of DMAc donor were also reported in literature (*Adv. Mater.* 2017, 29, 1701476). In the excited state, the twist geometry has lower singlet and triplet energies than the ground state. In the triplet excited state of TBP-1DMAc, the local minimum at around 0° and 90° are 2.67 eV and 2.40 eV. The large energy difference indicates that all conformations will distribute at around 90° according to Boltzmann distribution relationship. In the manuscript, we only consider the singlet and triplet excited state. The data were summarized in Supplementary Table 4.

Figure R5. Flexible PES scanning of the D-A dihedral angles of TBP-1DMAc at ground state and the Boltzmann distributions of the geometries with different dihedral angles in ground state.

Supplementary Table 4. TD-DFT calculated ΔE_{ST} in the ground state geometries (QA or QE conformer) and the average ΔE_{ST} according to Boltzmann distribution on the scanned S_1 and T_1 PES in Fig 2.

	ΔE_{ST} in S_0 geometry meV	S_1 average ΔE_{ST} ^{a)} meV	T_1 average ΔE_{ST} ^{a)} meV
TBP-1MCz	40.4	23.6	25.6
TBP-1DMAc	462 (QA 33%)/7.7 (QE 67%) ^{b)}	25.4	75.4
TBP-1aDMAc	463 (QA 94%)/8.5 (QE 6%) ^{b)}	28.7	153.8 (QA 11%/QE 89%)

a) The conformer distributions were estimated by Boltzmann distribution: $\%Conformer i =$

$$\frac{\exp(-\frac{E_i}{k_b T})}{\sum_j \exp(-\frac{E_j}{k_b T})}, \text{ where } E_i \text{ is the conformational energy of conformer } i \text{ calculated by DFT or TD-DFT,}$$

k_b is Boltzmann constant and T is the ambient temperature (298 K); and the average energies were calculated by timing TD-DFT calculated ΔE_{ST} in each geometry with the $\%Conformer i$ of the corresponding scanned singlet or triplet PES.

b) The ΔE_{ST} calculated in quasi-axial (QA) and quasi-equatorial (QE) conformation and the percentages of each conformation calculated by the geometry energy and Boltzmann distribution.

8. Inconsistency in naming of compounds have been noted.

Response: We are sorry for the ambiguity. The name of TBP-DMAc was from the previous work from our group so we used it directly (*Adv. Funct. Mater.* 2018, 28, 1704927). And the new compounds named TBP-3MCz and TBP-3aDMAc were to reflect the multiple donor character. In the potential surface scanning section, to simplify the calculation, we used the structure with only one donor attached to it. To void ambiguity, we named them as “TBP-1MCz”, “TBP-1DMAc” and “TBP-1aDMAc”. We hope that this naming can be better for reading.

9. Lines 469-470, “The transient lifetime measurements were conducted in vacuum.” Does the author mean the whole instrument is placed in a vacuum chamber or is the sample after degassing is stored under vacuum?

Response: In the lifetime measurement, the samples were placed in the chamber of an Oxford Instruments liquid nitrogen cryostat and then the chamber was vacuumized. The excitation and emission light can transmit through the window of the chamber to give PL or lifetime signals.

10. Line 500, does the author use oxygen or ozone to treat ITO glass?

Response: We use oxygen plasma to treat ITO glass.

11. Supplementary Line 67, “different conformers have different transition dipole moments”

Response: Thanks for your kind reminders. The typo has been corrected.

12. Supplementary synthesis section, Line 239, Pd(OAc)₂ is called palladium(II) acetate.

Response: We are sorry for the error. It has been corrected.

13. Captions for Supplementary Fig 17-20, ¹³C{¹H} NMR spectra and CDCl₃-d.

Response: The ¹H and ¹³C spectra were placed separately and the captions were modified in SI.

14. Could the author explain how the computational methods are chosen since over half of the article is based on computation? Have the authors done any benchmarking?

Response: Thanks for your question. In geometry optimizations, we used B3LYP functional which is the most general functional in TADF research and in combined with DFT-D3 correction, it can accurately optimize the molecular geometries with reduced cost (*J. Chem. Theory Comput.* 2015, 11, 4, 1481–1492). And as shown in the answer of question 3 from Reviewer #1, optimized geometry in B3LYP(D3)/def2SVP level shows negligible difference in compared with that optimized using ωB97XD functional. In TD-DFT calculations, the CT state is highly dependent on the Hartree–Fock exchange percentage in the exchange–correlation functional. For calculations of TADF materials, optimal Hartree–Fock percentage in the exchange–correlation functional method or tuned range-separated hybrid functionals can give reliable prediction on CT energies and ΔE_{ST} (*J. Chem. Theory Comput.* 2013, 9, 3872.; *J. Chem. Theory Comput.* 2015, 11, 3851). Therefore, we used tuned range-separated hybrid functional ωB97XD* for TD-DFT single point calculations. As suggested by reviewers, we compared the excited state energies of TBP-3MCz based on the same geometry. The range-tuned ωB97XD* functional results can accurately predict the singlet and triplet energies, while other functionals such as M062x and the un-tuned ωB97XD cannot give the precise energies. The B3LYP functional can also give the proper ΔE_{ST}, despite the slight underestimation of the excited state energies. Therefore, the calculation using B3LYP functional in the potential surface

scanning and TD-DFT calculation of the molecular dynamic results can also be representative. We also managed to compare the results with a double hybrid functional DSD-PBEP86-D3 using ORCA, which can describe the CT excitation with improved accuracy (*J. Chem. Theory Comput.* 2022, 18, 1646-1662). Due to the large computation cost (need memory >100 GB for TBP-3MCz), only a smaller def-TZVP basis set can be adopted. However, we cannot assess to the lowest triplet state after double correction of the initial generated triplet states. Nevertheless, the singlet energy is very close to the ω B97XD*/def2SVP result, which can verify the accuracy of our computation methods.

Supplementary Table 2. Comparison of the calculated excited state energies of TBP-3MCz calculated by different methods.

Methods	ω B97XD*/ def2-SVP	DSD-PBEP86/ def-TZVP ^{a)}	ω B97XD/ def2-SVP	B3LYP/ def2-SVP	M062x/ def2-SVP	PBE0/ def2-SVP	Exp. ^{b)}
S ₁ / eV	2.553	2.594	3.813	2.428	3.559	2.636	~2.60
T ₁ / eV	2.521	-	2.977	2.398	3.104	2.584	~2.55
ΔE_{ST} / eV	0.032	-	0.836	0.030	0.455	0.052	0.05

a) Considering the computational cost, only def-TZVP basis set can be adopted for the double hybrid functional calculation, and we cannot assess to the lowest triplet state after the double correction of the initial 10 triplet states in the calculation (output energy >8 eV); b) experimental energy levels in diluted toluene solution, estimated from the peak wavelength of the fluorescence and phosphorescence spectra measured at 77 K.

15. Please provide detailed description on how to measure those rates and also the fitting protocols.

Response: We use the build-in FL980 software to conduct the multiple exponential decay fitting. Here is the example of the fitting on the transient PL decay of 15 wt% TBP-3MCz: PhCzBCz film. The τ_1 is prompt lifetime and the delayed lifetimes are averaged from τ_2 and τ_3 . And the calculation of the rate constants was attached in Supporting Information in “Excited state lifetimes and rate constants calculation” section.

Figure R6. Fitting results of the transient decay PL decay of TBP-3MCz: PhCzBCz film.

As the explanations offered to explain the experimental findings are not concrete enough and the effect of conformation on TADF behaviour is not unprecedented in the literature, this article may not be suitable to be published in nature communication.

Response: In the revised manuscript, the explanations of the experimental findings were enhanced by addressing the questions and discussions with additional data. The influence of conformation distribution in host-guest system on the TADF lifetime was investigated by theoretical simulations, photophysical properties measurements and OLED characterizations, and the results can provide a new understanding of conformation distribution and excited state lifetime of TADF materials, which are different from the previous literatures. We believe that the revised article is within the scope of *Nature Communications*.

REVIEWER COMMENTS

Reviewer #1 (Remarks to the Author):

I fully agreed with the comment of Review #3 on the novelty of the present work by Qiu et al. In their work, effect of molecular conformations on TADF behavior has been well known in the literatures. Therefore, I still thought that their work did NOT provide new physical and chemical insight into TADF materials, although they comprehensively made experimental measurements and theoretical calculations. Therefore, I really cannot recommend it be published in Nature Communications with a high impact.

1. Regarding my Comment #2 in the last-round concerns, the authors stated that “However, “most of TADF emitters show k_{RISC} values of $> 1 \times 10^6 \text{ s}^{-1}$ ” is invalid. Only a few TADF materials can achieve fast spin-flipping among the reports on TADF materials developed in decades (k_{RISC} typically range from 103 to 10^5 s^{-1}), and most of them have been cited in the introduction.” However, obviously, the authors did not check the progress in RISC rates for TADF molecules. As collected by Kaji and co-workers (Supplementary Table 4 in Nat. Photonics 14, 643–649 (2020).), RISC rates of many TADF molecules had gone beyond 10^6 s^{-1} by 2019. Recent three years, there are more works reported with k_{RISC} of over 10^6 s^{-1} , for example: Nature Photonics 14, pages 636–642 (2020); Angew. Chem. Int. Ed. e202210210 (2022); Adv. Optical Mater. 2102339 (2022); J. Mater. Chem. C, 10, 1313–1325 (2022). Therefore, I cannot understand what progress the present work made!

2. The authors provided vacuum-evaporated OLED results; it is very helpful to evaluate the material performances. Optimization of solution processed OLED performances are more complicated; they are not so stable as evaporated devices. Based on the results, I further convinced that not only the chemical structures but also their material performances are not outstanding as they claimed. Evaporated OLEDs with similar or even better performances have often noticed. It means good efficiency roll-off is more likely because of their detailed solution process rather than their emitter materials.

3. The authors claimed evaporated devices showed much better maximum EQE than solution processed ones because of the high horizontal dipole moment that can help to enhance outcoupling. However, they didn't provide the corresponding horizontal dipole moment results using solution process. The solution process films have a large chance to show higher horizontal dipole moment because of centrifugal force during spin coating. Assuming they outcoupling are similar for evaporated and solution processed OLEDs, it is more likely the roll-off is underestimated because they didn't get the maximum EQE.

4. As I previously mentioned, the authors didn't provide enough experimental evidence for the different conformations and conformational distributions. Only one crystal result of QA cannot support their claims. Conformational distributions based on Boltzmann distribution cannot accurately evaluate the results in complicated environment as amorphous states.

Reviewer #2 (Remarks to the Author):

Overall, the authors have made significant improvements to the manuscript and given detailed responses to my comments. However, before the manuscript can be accepted I recommend further (minor) revision, to include more of the response text into the manuscript and SI. Relevant information that strengthens the manuscript should not be restricted to the authors rebuttal letter.

Specifically:

Reviewer 2, point 1. The specific response text stating "...the previous investigation on dual conformation pays less attention on the excited state lifetime and RISC process....." is a good point and should be incorporated into the manuscript.

Reviewer 2, point 3. The response text and Figure R2 should be included in SI.

Reviewer #3 (Remarks to the Author):

This manuscript is ready to be published in Nature Communications, provided that the issue below are addressed.

1. Line 24, The term 'partial conformer distribution' is even more confusing and language-wise is incorrect. Would it better to say " Acridine-type flexible donors have a broad conformation distribution or bimodal distribution, in which some conformers feature large singlet-triplet energy gap, leading to long excited state lifetime."?

2. I think the authors are a bit confused on the term "transition dipole moment" and "dipole moment". The former refers to the dipole moment associated with transition between two states, while the latter describes the uneven distribution of charges on the molecules. Please correct all these in the manuscript and supporting information.

Point-by-point responses to the reviewers' comments and suggestions:

Reviewer #1 (Remarks to the Author):

I fully agreed with the comment of Review #3 on the novelty of the present work by Qiu et al. In their work, effect of molecular conformations on TADF behavior has been well known in the literatures. Therefore, I still thought that their work did NOT provide new physical and chemical insight into TADF materials, although they comprehensively made experimental measurements and theoretical calculations. Therefore, I really cannot recommend it be published in Nature Communications with a high impact.

Response: Thanks for your time and precious suggestions for improvement of our manuscript. The results have been strengthened by addressing the comments and the novelty of our work has been elucidated in the introduction section. We believe our work can shade light on the complicated conformation distribution in amorphous host-guest system and provide concrete guideline for future TADF materials development. As the modification of the original manuscript was recognized by other reviewers, we would be grateful if the reviewer can reconsider our manuscript and provide valuable suggestions.

1. Regarding my Comment #2 in the last-round concerns, the authors stated that “However, “most of TADF emitters show k_{RISC} values of $> 1 \times 10^6 \text{ s}^{-1}$ ” is invalid. Only a few TADF materials can achieve fast spin-flipping among the reports on TADF materials developed in decades (k_{RISC} typically range from 10^3 to 10^5 s^{-1}), and most of them have been cited in the introduction.” However, obviously, the authors did not check the progress in RISC rates for TADF molecules. As collected by Kaji and co-workers (Supplementary Table 4 in Nat. Photonics 14, 643–649 (2020).), RISC rates of many TADF molecules had gone beyond 10^6 s^{-1} by 2019. Recent three years, there are more works reported with k_{RISC} of over 10^6 s^{-1} , for example: Nature Photonics 14, pages 636–642 (2020); Angew. Chem. Int. Ed. e202210210 (2022); Adv. Optical Mater. 2102339 (2022); J. Mater. Chem. C, 10, 1313–1325 (2022). Therefore, I cannot understand what progress the present work made!

Response: Thanks for your comments. We totally comprehended the progress in the RISC rate, and we have made the statement that “the excited state lifetime of the TADF emitters can be minimized to a few microseconds or sub-microsecond time-range, corresponding to the rate constant of RISC (k_{RISC}) greater than 10^6 s^{-1} ” in the second paragraph in the introduction section. The representative literatures reporting high k_{RISC} values have been referenced correctly as reference 11, 16, 19 and 24–33. However, instead of reporting record high k_{RISC} values, the progress in our manuscript mainly lies in investigating the influence of conformation distribution on the RISC process of D-A type TADF molecules comprehensively from theoretical simulation and photo-physical measurement to OLED device characterizations. We hope to correlate the molecular structure with the excited state lifetime in practical amorphous film state with conformation distribution. And the molecular design of confining conformation for efficient RISC process can be proposed according to our results, and this strategy was further verified by newly developed prototype TADF emitters. We believe there are no existing overlapped literatures involving the similar investigations. Moreover, the previous works on fast RISC rate mainly focus on the electronic structures of isolated molecules such as energy level, coupling with LE state and SOC effect, and neglect the conformation distribution in amorphous film. For example, the TRZCzPh-BNCz molecule in the listed literature Angew. Chem. Int. Ed. e202210210 (2022) obtained fast k_{RISC} in solution state without considering the

conformation distribution effect in amorphous film. Since the effect of conformation distribution really exists and is important in TADF-OLED but neglected by many researches, we believe our work can arouse the attention to conformation distribution in amorphous state when designing novel TADF materials with short excited state lifetime. The introduction section in the manuscript was modified to better demonstrate our novelty.

2. The authors provided vacuum-evaporated OLED results; it is very helpful to evaluate the material performances. Optimization of solution processed OLED performances are more complicated; they are not so stable as evaporated devices. Based on the results, I further convinced that not only the chemical structures but also their material performances are not outstanding as they claimed. Evaporated OLEDs with similar or even better performances have often noticed. It means good efficiency roll-off is more likely because of their detailed solution process rather than their emitter materials.

Response: Solution-processed OLED with the advantages of low material consumption and low-cost processing is advantageous for large-area device fabrication, which can significantly reduce the price of the OLED in practical applications (*Adv. Mater.*, 2022, e2207454). Therefore, developing highly efficient solution-processed OLED is meaningful in both academia and industry but challenging. That is the reason for the need for progress in solution-processed TADF OLED. We demonstrated efficient solution-processed OLED with high efficiency and excellent efficiency roll-off suppression by conformation confinement strategy, which contributes to the development of solution-processed OLED and would be of interest for the readers in this field. Moreover, the solution-processed OLED in our work has a very simple device structure (ITO/PEDOT: PSS/EML/TmPyPb/CsF/Al), and the fabrication processes are general processes, which are not complicated especially in compared with the vacuum-evaporated ones. The fabrication processes are easeful and time-saving. With such a simple device structure, similar device performance as the vacuum-evaporated OLED can be obtained. We think it is one of the merits of these materials.

In addition, in the reference vacuum-evaporated OLEDs we fabricated (Supplementary Figure 22), we used the same PhCzBCz host for consistency without host optimization. CT-type TADF emitters in nonpolar carbazole-based host generally shows lower device efficiency in compared with the polar host such as DPEPO in vacuum-evaporated OLED (*Adv. Sci.*, 2021, e2102141 and *Adv. Opt. Mater.* 2021, 2101343). Therefore, the full potential of the material in vacuum-evaporated device might not be released using PhCzBCz host. But our manuscript does not aim at achieving top efficiency but investigate the structure-property relationship of conformation distribution. Therefore, we believe the current OLED device characterizations can demonstrate the success of the proposed molecular design strategy.

3. The authors claimed evaporated devices showed much better maximum EQE than solution processed ones because of the high horizontal dipole moment that can help to enhance outcoupling. However, they didn't provide the corresponding horizontal dipole moment results using solution process. The solution process films have a large chance to show higher horizontal dipole moment because of centrifugal force during spin coating. Assuming they outcoupling are similar for evaporated and solution processed OLEDs, it is more likely the roll-off is underestimated because they didn't get the maximum EQE.

Response: Thanks for your suggestions. Molecular orientation in solution-processed OLED is a very important topic to further improve the device efficiency. However, achieving high transition dipole orientation by solution process is in fact challenging. Currently, only one stick-like heptafluorene fluorescence molecule was reported to have high orientation in solution-processed OLED (*Applied Physics Letters*, 2015, 106, 063301). In solution-processed TADF-OLEDs, only a few literatures report molecular orientation, and the highest one only shows a horizontal orientation ratio of 73% (*Nat. Commun.*, 2022, 13, 7828; *Angew. Chem. Int. Ed.*, 2022, 61, e202212861; *Angew. Chem. Int. Ed.*, 2023, e202218911). Given that many organic photovoltage materials show good molecular stacking in the solution-processed film (*Joule* 2019, 3, 1140-1151; *Nat. Commun.* 2021, 12, 332), we think achieving high horizontal transition dipole orientation in solution-processed TADF-OLED is possible but special molecular design and process methods are needed, and we are also conducting some investigations on this topic in our lab.

We measured the angular-dependence of *p*-polarized light intensity of the solution-processed films and simulated the transition dipole orientation (Supplementary Figure 23). Different from the vacuum-evaporated film, the spin-coated film shows no horizontal transition dipole orientation. The 15 wt% TB-3MCz: PhCzBCz film and 15 wt% TBP-3MCz: PhCzBCz film have a horizontal transition dipole orientation ratio of 67%, indicating an isotropic character, while the 15 wt% TRZ-3MCz: PhCzBCz film has slightly vertical orientation. This indicates that the solution-processed OLEDs have an ordinary light-outcoupling. Moreover, we also conducted optical simulation on the 15 wt% TBP-3MCz: PhCzBCz based solution-processed OLED device and the out-coupled fraction of 26.2% was obtained. Therefore, the obtained maximum EQE of 24.4% for TRZ-3MCz based device is reasonable considering the PLQY. We believe the efficiency roll-off was not underestimated. It is interesting to note that in the solution-processed OLED with no molecular orientation, the fraction of evanescent field is large, which is originated from the surface plasmon resonance. This loss can be reduced by horizontal orientation of the transition dipole moments (*Physical Review Applied*, 2017, 8, 037001). Therefore, improving the molecular orientation in the solution-processed film is an important topic for future development. The discussion on light outcoupling was supplemented as Supplemental Figure 23 in supporting information.

Supplemental Figure 23. Evaluation of the horizontal orientations of the transition dipole moments in solution-processed and vacuum-evaporated emission layers. Angular-dependent of *p*-polarized PL intensities of the vacuum-evaporated a) 15 wt% TRZ-3MCz: PhCzBCz and b) 15 wt% TB-3MCz: PhCzBCz films and solution-processed c) 15 wt% TRZ-3MCz: PhCzBCz, b) 15 wt% TB-3MCz: PhCzBCz and e) 15 wt% TBP-3MCz: PhCzBCz films and the fitted horizontal ratios of the transition dipole moments; f) Simulated power dissipation to the different optical modes in the solution-processed OLED (stack structure: ITO/PEDOT: PSS (40 nm)/ 15 wt% TRZ-3MCz: PhCzBCz (30 nm)/ TmPyPb /CsF /Al).

4. As I previously mentioned, the authors didn't provide enough experimental evidence for the different conformations and conformational distributions. Only one crystal result of QA cannot support their claims. Conformational distributions based on Boltzmann distribution cannot accurately evaluate the results in complicated environment as amorphous states.

Response: Thanks for your comments. We had investigated the conformation distributions in host-guest systems theoretically by DFT calculations and MD simulations of the host-guest system, and experimentally by time-resolved PL and EL spectra. As CT emission is originated from the transition from donor to acceptor, which is sensitive to the relative positions of the donor and acceptor moieties. In the amorphous film with conformational distributions, there will be energy

distributions of CT state, which can be reflected by the change of the spectra. Therefore, in the time-resolved PL and EL spectra, the spectral shift as delayed time in the time-resolved spectra in TBP-3aDMAc clearly indicates the broad conformation distribution effect, while TBP-3MCz with confined conformation distribution shows negligible spectral change (Figure 4). Moreover, in the previous revised manuscript, following the suggestions from the Reviewer #1, the QA conformation and energy transfer processes of TBP-3aDMAc were discussed by single crystal result and additional photo-physical characterizations (Supplementary Figure 3). The discussion on triplet energy transfer effect between conformers (Supplementary Figure 12), which can be found in the answer of the question 9 from Reviewer #1 in the first point-by-point response and “time-resolved spectra analyses for conformation distribution” section in the manuscript, can also illustrate the triplet energy distribution effect in the amorphous film originated from conformer distributions.

In addition, the transient lifetime decay curve also contains the information of conformation distribution. A settled conformer has settled emission transient decay lifetime, showing double exponential decay with one prompt fluorescence component and delayed fluorescence component:

$$I(t) = A_{PF}e^{-\frac{t}{\tau_{PF}}} + A_{DF}e^{-\frac{t}{\tau_{DF}}}$$

In the amorphous film state, the broad conformation distribution would lead to multiple exponential decay character, as the different transient decay character of the sample in diluted degassed toluene solution (Supplementary Figure 15c) and amorphous film state (Figure 5c). The transient PL decay of the delayed fluorescence in amorphous film state is the integration of the multiple single exponential decay from different conformers with different lifetime, and can be represented as:

$$I_{DF}(t) = \sum_i A_i e^{-\frac{t}{\tau_i}}$$

If we use multiple specific τ values (covering the whole lifetime region) to fit the transient decay curve, the obtained amplitude values (A_j) from the fitting result can represent the contribution of the component with the given lifetime. The percentage of each component with the given lifetime (p_j) can be calculated as:

$$p_j = A_j \tau_j / \sum_i A_i \tau_i$$

Plotting the p_j against lifetime, the distribution of delayed lifetime can be obtained (Supplementary Figure S13).

As can be seen in the delayed lifetime distribution, the 15 wt% TBP-3MCz: PhCzBCz system with the shortest excited state lifetime also shows a narrow lifetime distribution, while the 15 wt% TBP-3aDMAc: PhCzBCz has a broad delayed lifetime distribution. The conformation distribution, especially the dihedral angle distribution in the amorphous film state, differentiates the RISC process in different molecules, causing a broad distribution of delayed fluorescence lifetime. TBP-3MCz with confined conformation distributions results in narrow and short delayed lifetime distributions. Combining the energy distributions (time-resolved spectra in Figure 4) and delayed lifetime distributions (Supplementary Figure S13), we believe that the conformation distribution of the emitters in amorphous state can be revealed experimentally. And the conclusion that the confined donor conformation for efficient TADF with short delayed lifetime can be drawn. The additional evidence and discussion were added to the manuscript and Supplementary Figure 13.

Supplementary Figure 13. Fitted transient PL decay curves (with R-Square > 0.995) and Delayed lifetime distribution of the 15 wt% TBP-3MCz, TBP-DMAc and TBP-3aDMAc doped PhCzBCz films. The τ_j are given parameters in multiple exponential decay fitting and A_j are fitted parameters.

Reviewer #2 (Remarks to the Author):

Overall, the authors have made significant improvements to the manuscript and given detailed responses to my comments. However, before the manuscript can be accepted I recommend further (minor) revision, to include more of the response text into the manuscript and SI. Relevant information that strengthens the manuscript should not be restricted to the authors rebuttal letter.

Specifically:

Response: Thanks for the positive comments on our revised manuscript.

Reviewer 2, point 1. The specific response text stating “...the previous investigation on dual conformation pays less attention on the excited state lifetime and RISC process...” is a good point and should be incorporated into the manuscript.

Response: We appreciate the suggestion. The statement was incorporated into the introduction to better demonstrate the distinguishment of our work from the previous investigation on dual conformation phenomenon.

Reviewer 2, point 3. The response text and Figure R2 should be included in SI.

Response: Thanks for the suggestion. The Figure R2 and the corresponding response have been attached to SI as Supplementary Figure 20.

Reviewer #3 (Remarks to the Author):

This manuscript is ready to be published in Nature Communications, provided that the issue below are addressed.

Response: Thanks for the recognition of our revised manuscript.

1. Line 24, The term ‘partial conformer distribution’ is even more confusing and language-wise is incorrect. Would it better to say “Acridine-type flexible donors have a broad conformation distribution or bimodal distribution, in which some conformers feature large singlet-triplet energy gap, leading to long excited state lifetime.”?

Response: Many thanks for your suggestion. We have modified the abstract according to the suggestion, and the term in the manuscript has also been changed for clarity.

2. I think the authors are a bit confused on the term “transition dipole moment” and “dipole moment”. The former refers to the dipole moment associated with transition between two states, while the latter describes the uneven distribution of charges on the molecules. Please correct all these in the manuscript and supporting information.

Response: We are sorry for the mistakes. It should be “transition dipole moment”. We had changed the original term “horizontal dipole orientation” to “horizontal orientation of the transition dipole moments” in the revised manuscript.